# AMPK activation counteracts cardiac hypertrophy by reducing O-GlcNAcylation

Roselle Gélinas[1], Florence Mailleux[1], Justine Dontaine[1], Laurent Bultot[1], Bénédicte Demeulder[1], Audrey Ginion[1], Evangelos P. Daskalopoulos[1], Hrag Esfahani[2], Emilie Dubois-Deruy[2], Benjamin Lauzier[3], Chantal Gauthier[3], Aaron K. Olson[4,5], Bertrand Bouchard[5], Christine Des Rosiers[5,6], Benoit Viollet[7,8,9], Kei Sakamoto[10], Jean-Luc Balligand[2], Jean-Louis Vanoverschelde[1,11], Christophe Beauloye[1,11], Sandrine Horman[1] & Luc Bertrand[1]

AMP-activated protein kinase (AMPK) has been shown to inhibit cardiac hypertrophy. Here, we show that submaximal AMPK activation blocks cardiomyocyte hypertrophy without affecting downstream targets previously suggested to be involved, such as p70 ribosomal S6 protein kinase, calcineurin/nuclear factor of activated T cells (NFAT) and extracellular signal-regulated kinases. Instead, cardiomyocyte hypertrophy is accompanied by increased protein O-GlcNAcylation, which is reversed by AMPK activation. Decreasing O-GlcNAcylation by inhibitors of the glutamine:fructose-6-phosphate aminotransferase (GFAT), blocks cardio-myocyte hypertrophy, mimicking AMPK activation. Conversely, O-GlcNAcylation-inducing agents counteract the anti-hypertrophic effect of AMPK. In vivo, AMPK activation prevents myocardial hypertrophy and the concomitant rise of O-GlcNAcylation in wild-type but not in AMPKα2-deficient mice. Treatment of wild-type mice with O-GlcNAcylation-inducing agents reverses AMPK action. Finally, we demonstrate that AMPK inhibits O-GlcNAcylation by mainly controlling GFAT phosphorylation, thereby reducing O-GlcNAcylation of proteins such as troponin T. We conclude that AMPK activation prevents cardiac hypertrophy predominantly by inhibiting O-GlcNAcylation.

[1] Pole of Cardiovascular Research, Institut de Recherche Expérimentale et Clinique, Université catholique de Louvain, Brussels, 1200, Belgium. [2] Pole of Pharmacotherapy, Institut de Recherche Expérimentale et Clinique, Université catholique de Louvain, Brussels, 1200, Belgium. [3] l'institut du thorax, INSERM, CNRS, Univ. Nantes, Nantes, 44007, France. [4] Department of Pediatrics, University of Washington School of Medicine and Seattle Children's Research Institute, Seattle, 98105-0371 WA, USA. [5] Montreal Heart Institute, Montreal, H1T 1C8, Canada. [6] Department of Nutrition, Université de Montréal, Montreal, H3T 1A8, Canada. [7] Institut Cochin, INSERM U1016, 75014 Paris, France. [8] CNRS UMR8104, 75014 Paris, France. [9] Université Paris Descartes Sorbonne Paris Cité, Paris, 75014, France. [10] Nestlé Institute of Health Sciences SA, Lausanne, 1015, Switzerland. [11] Division of Cardiology Cliniques Universitaires Saint-Luc, Brussels, 1200, Belgium. Roselle Gélinas and Florence Mailleux contributed equally to this work. Correspondence and requests for materials should be addressed to L.B. (email: luc.bertrand@uclouvain.be)

Pathological cardiac hypertrophy is considered to be a compensatory mechanism that aims to maintain cardiac function in the face of mechanical or neurohormonal stresses[1]. However, sustained pathological cardiac hypertrophy eventually becomes maladaptive and is a predictor of heart failure[2]. Intracellular signaling pathways regulating cardiac hypertrophy development are multiple and complex[3, 4]. Amongst them, the most extensively studied ones are the calcineurin/nuclear factor of activated T cells (NFAT) and mitogen-activated protein kinase ERK pathways promoting gene expression, as well as the mammalian target of rapamycin (mTOR)/p70 ribosomal S6 protein kinase (p70S6K) and eukaryotic elongation factor-2 (eEF2) pathways involved in the stimulation of protein synthesis[3, 5].

AMP-activated protein kinase (AMPK) is a cellular fuel gauge, which can detect energetic disequilibrium occurring under metabolic stress[6, 7]. Once activated, AMPK inhibits various anabolic pathways, including protein synthesis via its action on both mTOR/p70S6K and eEF2 pathways[8, 9], and enhances catabolic pathways, such as glycolysis, to restore energetic balance required for cell survival[7, 10]. Because of its dampening action on protein synthesis, AMPK has been suggested to be a putative inhibitor of cardiac hypertrophy. In line with this interpretation, AMPK activation by activators such as 5-Aminoimidazole-4-carboxamide ribonucleoside (AICAr), metformin or resveratrol prevents hypertrophy induced by phenylephrine (PE) in cultured cardiomyocytes[11, 12]. This not only correlates with alteration of p70S6K and eEF2 phosphorylation and decrease in protein synthesis, but also with inhibition of ERK and NFAT signaling[11, 13, 14]. Likewise, AMPK activation by AICAr, metformin or adiponectin attenuates cardiac hypertrophy and improves cardiac function in rodent models subjected to transverse aortic constriction (TAC) or isoproterenol treatment, and this is concomitant with inhibition of the afore-mentioned signaling pathways[13, 15–17]. However, there is no robust evidence demonstrating that all these downstream signaling pathways are involved in the anti-hypertrophic action of AMPK.

O-linked N-acetylglucosamine (O-GlcNAc) is a post-translational protein modification occurring on Ser/Thr residues. A small but significant part of cellular glucose enters the hexosamine biosynthesis pathway (HBP), under the control of glutamine:fructose-6-phosphate aminotransferase (GFAT), finally producing UDP-GlcNAc, which then serves as substrate for O-GlcNAcylation. Besides GFAT, O-GlcNAcylation is regulated by two other enzymes, O-GlcNAc transferase (OGT) and β-N-acetylglucosaminidase (OGA)[18]. OGT adds and OGA removes the O-GlcNAc moiety, respectively[18]. HBP is involved in multiple physiological processes but is also associated with undesirable cellular events in chronic diseases, such as diabetes inducing adverse effects in the heart[18, 19]. In relation to cardiac pathologies, O-GlcNAcylation levels are increased during acute myocardial ischemia and chronic heart failure, but in these cases, with a cardioprotective effect[18, 20, 21]. The role of O-GlcNAc during cardiac hypertrophy development is complex and still remains partly unclear[18, 21]. Action of O-GlcNAc largely depends on the context of cardiac hypertrophy with distinctive roles in hypertrophy development when linked to diabetes or to physiological exercise or to pressure overload pathological conditions[18, 21]. Regarding our topic, cardiac O-GlcNAc signaling and O-GlcNAcylation levels are increased in rats with pressure overload-mediated cardiac hypertrophy and in patients with aortic stenosis[22, 23]. Similarly, O-GlcNAc is increased in neonatal rat ventricular myocytes (NRVMs) submitted to pro-hypertrophic stimuli, and pharmacological inhibition of O-GlcNAc signaling reverses the hypertrophic transcriptional reprogramming[23].

The present study was undertaken to better define the inhibitory role of AMPK in pathological cardiac hypertrophy development and to unambiguously identify the key cellular events involved in this process. Using low concentrations of AMPK activators, including the direct activator A769662[24], we show that AMPK activation efficiently inhibits cardiomyocyte hypertrophy without affecting any of the previously-described AMPK downstream targets, suggesting that AMPK regulates cardiac hypertrophy via a not-yet-identified mechanism. Inasmuch as AMPK is a known regulator of glucose metabolism[7, 10], we sought potential links between AMPK, cardiac hypertrophy prevention and O-GlcNAcylation process. Here, we report that an increase in protein O-GlcNAcylation is required for cardiac hypertrophy development. More importantly, we demonstrate that AMPK activation prevents both cardiomyocyte hypertrophy in vitro and cardiac hypertrophy in vivo by inhibiting O-GlcNAc signaling via its actions on GFAT and OGT.

Taken together, our results demonstrate that AMPK activation prevents both in vitro and in vivo cardiac hypertrophy development predominantly by decreasing protein O-GlcNAcylation.

## Results

**AMPK activation by A769662 prevents NRVM hypertrophy.** First, we assessed the ability of A769662, a selective and direct allosteric activator of AMPK[25] to activate the AMPK pathway in NRVMs. We started by using a concentration of 100 μM giving maximal AMPK activation in adult cardiomyocytes[26]. We showed that, at this concentration, A769662 rapidly increased AMPK activity, which was maximal at 5 min and persisted for at least 24 h (Supplementary Fig. 1a). The same applied for the phosphorylation of its bona fide substrate acetyl-CoA carboxylase (ACC) (Supplementary Fig. 1b).

Next, we verified the capacity of A769662 to prevent PE-induced NRVM hypertrophy. NRVMs were treated for 24 h with 20 μM PE to promote hypertrophy[12, 27], in the presence or absence of 100 μM A769662 (Supplementary Fig. 1c). A769662-mediated phosphorylation of AMPK and of ACC was similar in cells treated with or without PE (Supplementary Fig. 1d, e). PE induced a significant increase in cell size (~2-fold, $p < 0.05$ two-way ANOVA with bonferroni post-test), as revealed by α-actinin immunostaining (Supplementary Fig. 1f, g), which was abolished by A769662 treatment. Moreover, mRNA level of *Nppb*, a genetic marker of pathological cardiac hypertrophy[28], was significantly elevated by PE (~2-fold, $p < 0.05$, two-way ANOVA with bonferroni post-test), this increase being prevented by A769662 (Supplementary Fig. 1h).

We next examined AMPK downstream targets previously proposed to be involved in the context of cardiac hypertrophy and we first analyzed p70S6K and eEF2, which are involved in the regulation of protein synthesis (Supplementary Fig. 1i-k). As reported by others[12, 29], PE increased p70S6K phosphorylation at its activating site Thr389 and decreased eEF2 phosphorylation at its inhibitory site Thr56. A769662 counteracted the action of PE on phosphorylation of both p70S6K and eEF2. In agreement, PE-induced cardiomyocyte hypertrophy was accompanied by a 2-fold increase in protein synthesis (monitored by radiolabeled amino acid incorporation into proteins), which was abrogated by co-treatment with 100 μM A769662 (Supplementary Fig. 1l). Regarding pathways regulating gene expression, PE induced the phosphorylation of both ERK1 and ERK2 isoforms at their dual-activating phosphorylation sites Thr202/Tyr204[30], and this was inhibited by A769662 (Supplementary Fig. 1i, m). Next, we examined the calcineurin/NFAT pathway by measuring NFAT transcriptional activity with a luciferase reporter gene[31]. PE treatment elicited a 3-fold to 4-fold increase in NFAT

transcriptional activity, which was fully inhibited by A769662 (Supplementary Fig. 1n). Because NFAT activation results in its translocation to the nucleus[31], we examined the cellular localization of NFATc3, one of the NFAT isoforms involved in the hypertrophic signal downstream of calcineurin[32]. NFATc3 was translocated to the nucleus upon PE stimulation and this was blocked by A769662 (Supplementary Fig. 1o, p).

**AMPK prevents hypertrophy independently of known targets.** To rule out a possible AMPK-independent action of A769662, as previously reported at high concentrations[33, 34], dose-response curves of A769662 were performed (Fig. 1a–h). Even at the lowest concentration used (12.5 μM), A769662 stimulated the AMPK signaling pathway (Fig. 1a) and fully inhibited PE-induced hypertrophy (Fig. 1b, c). To confirm AMPK involvement in A769662's action at this low concentration, both AMPKα1 and AMPKα2 catalytic subunits were knocked-down with small interfering RNA (siRNA) transfection (Fig. 1i–l). The 95% decrease in expression of AMPKα catalytic subunits (Fig. 1i) was accompanied by loss of A769662 action on AMPK signaling (Fig. 1j). This AMPK deletion, which was similar in control and PE-treated cells (Supplementary Fig. 2), abolished the anti-hypertrophic action of A769662 (Fig. 1k, l). This confirmed that A769662-mediated inhibition of NRVM hypertrophy is an AMPK-dependent mechanism.

Unexpectedly, none of the proposed AMPK targets was altered at low concentration (Fig. 1d–h). Indeed, while 12.5 μM A769662 fully curtailed cardiomyocyte hypertrophy, no inhibition of NFAT activity, ERK, and p70S6K phosphorylation, as well as amino acid incorporation into proteins could be observed at this concentration. Interestingly, this observation also occurred with other pro-hypertrophic stimulus and AMPK activators. Indeed, A769662-mediated inhibition of hypertrophy occurred without modification of known AMPK targets when Angiotensin II (AngII) (100 nM) was used as pro-hypertrophic agent (Supplementary Fig. 3). Besides, we chose to utilize AICAr and a member of the biguanide family, phenformin as other AMPK activators, such compounds having been used in the past at high concentration to inhibit hypertrophy[11, 12]. Similar to what was observed with A769662, AICAr (Fig. 2a–e and Supplementary Fig. 4a-c) and phenformin (Fig. 2f–j and Supplementary Fig. 4d–f) abolished PE-induced cardiomyocyte hypertrophy at concentrations ranging from 0.25 to 0.03 mM, respectively. Only higher doses of these compounds were able to block some of p70S6K, ERK, NFAT signaling and protein synthesis, whereas lower effective doses were unable to do so. Collectively, these results support the idea that low doses of AMPK activators are capable of preventing cardiomyocyte hypertrophy without acting on the aforementioned AMPK targets, suggesting the involvement of another mechanism.

**AMPK activation decreases protein O-GlcNAcylation.** Since HBP stimulation and O-GlcNAcylation (Fig. 3a), are known to occur during cardiac hypertrophy development[22], we assessed O-GlcNAc levels in NRVMs treated with PE. O-GlcNAc levels were rapidly increased following PE supplementation (before 2 h of PE treatment) (Fig. 3b, c). As previously shown in aortic banding induced-cardiac hypertrophy[22], this increase in O-GlcNAc levels correlated with an increase in protein level of GFAT, the rate-limiting step of HBP (Fig. 3b, d). This was rapidly followed by an increase in cell size (Fig. 3e), suggesting a link between O-GlcNAc levels and hypertrophy development.

Remarkably, AMPK activation by 12.5 μM A769662 blunted the increase in O-GlcNAc levels that occurs 24 h after PE treatment (Fig. 3f–i). Inasmuch as it has been recently shown that

AMPK can phosphorylate GFAT on Ser243 inducing its inactivation[35], we evaluated the impact of 12.5 μM A769662 on GFAT Ser243 phosphorylation and found a significant increase after 1 h of A769662 incubation (Fig. 3j). In agreement with the partial inhibition mediated by AMPK on GFAT[35], the effect of A769662 on cardiomyocyte size and O-GlcNAc levels were only observable after 8 h of treatment (Fig. 3k–l). We can reasonably hypothesize that this delay is due to the time necessary for GFAT inhibition to blunt HBP pathway. Nevertheless, those results show a perfect correlation between cardiomyocyte size and O-GlcNAc levels, reinforcing the link between O-GlcNAc increased and hypertrophy development.

We can conclude from these observations that hypertrophy is accompanied by an increase in GFAT expression promoting O-GlcNAcylation and that concomitant AMPK activation promotes GFAT phosphorylation leading to HBP inhibition and O-GlcNAc level reduction.

**O-GlcNAc inducers block the anti-hypertrophic effect of AMPK.** We then evaluated the putative key role of HBP/O-GlcNAc inhibition in the anti-hypertrophic action of AMPK by using O-GlcNAc inducers. Two different ways are commonly used to increase cellular O-GlcNAc levels (Fig. 3a). The rate-limiting HBP enzyme GFAT can be bypassed by feeding glucosamine (GlcN) to cells[18]. Alternatively, OGA can be inhibited pharmacologically by compounds such as O-(2-acetamido-2-deoxy-D-glucopyranosylidene)amino N-phenyl carbamate (PUGNAc)[18]. Both strategies have been used in our in vitro model. GlcN efficiently increased O-GlcNAc levels under basal condition (2-fold and 4-fold increase for PE and GlcN treatments, respectively, $p \leq 0.05$ vs. control, Student's $t$-test) and, more importantly, prevented the reduction in O-GlcNAc levels induced by A769662 under PE-treatment ($75 \pm 14\%$ for PE + A769662 vs. $140 \pm 60\%$ for PE + A769662 + GlcN, relative to PE treatment) (Fig. 4a). This occurred without affecting AMPK signaling (assessed by measuring ACC phosphorylation, which is increased 2-fold by A769662 in the presence or absence of GlcN) (Fig. 4a). Interestingly, GlcN had no effect on cell size when incubated alone showing that the increase in O-GlcNAc levels is not sufficient to induce cardiac hypertrophy by itself (Fig. 4b). However, when treated in combination with PE and A769662, GlcN remarkably blunted the anti-hypertrophic effect of the AMPK activator (Fig. 4b). The same applied to the two other AMPK activators, phenformin and AICAr demonstrating no dependency to the AMPK activator used (Fig. 4b).

GFAT inhibition by AMPK being partial[35], such inhibition should not fully blunt HBP flux. In agreement with this hypothesis, OGA inhibition by PUGNAc was able to increase O-GlcNAc levels even in the presence of A769662 ($78 \pm 22\%$ for PE + A769662 vs. $247 \pm 98\%$ for PE + A769662 + PUGNAc, relative to PE treatment, $p \leq 0.05$, Student's $t$-test) (Fig. 4c). Supporting data obtained with GlcN, PUGNAc nicely counteracted the anti-hypertrophic action of the three AMPK activators (Fig. 4d).

Next, we examined whether pharmacological HBP inhibition can recapitulate AMPK action and block cardiac hypertrophy development. Strikingly, 6-diazo-5-oxo-norleucine (DON) and Azaserine (Aza), two GFAT inhibitors known to reduce O-GlcNAc levels, blunted PE-induced hypertrophy, perfectly mimicking AMPK activators (Fig. 4e). GlcN treatment prevented the anti-hypertrophic action of DON and Aza, indicating that their anti-hypertrophic actions were clearly related to O-GlcNAc and that these events played an important role in cardiac hypertrophy development (Fig. 4e).

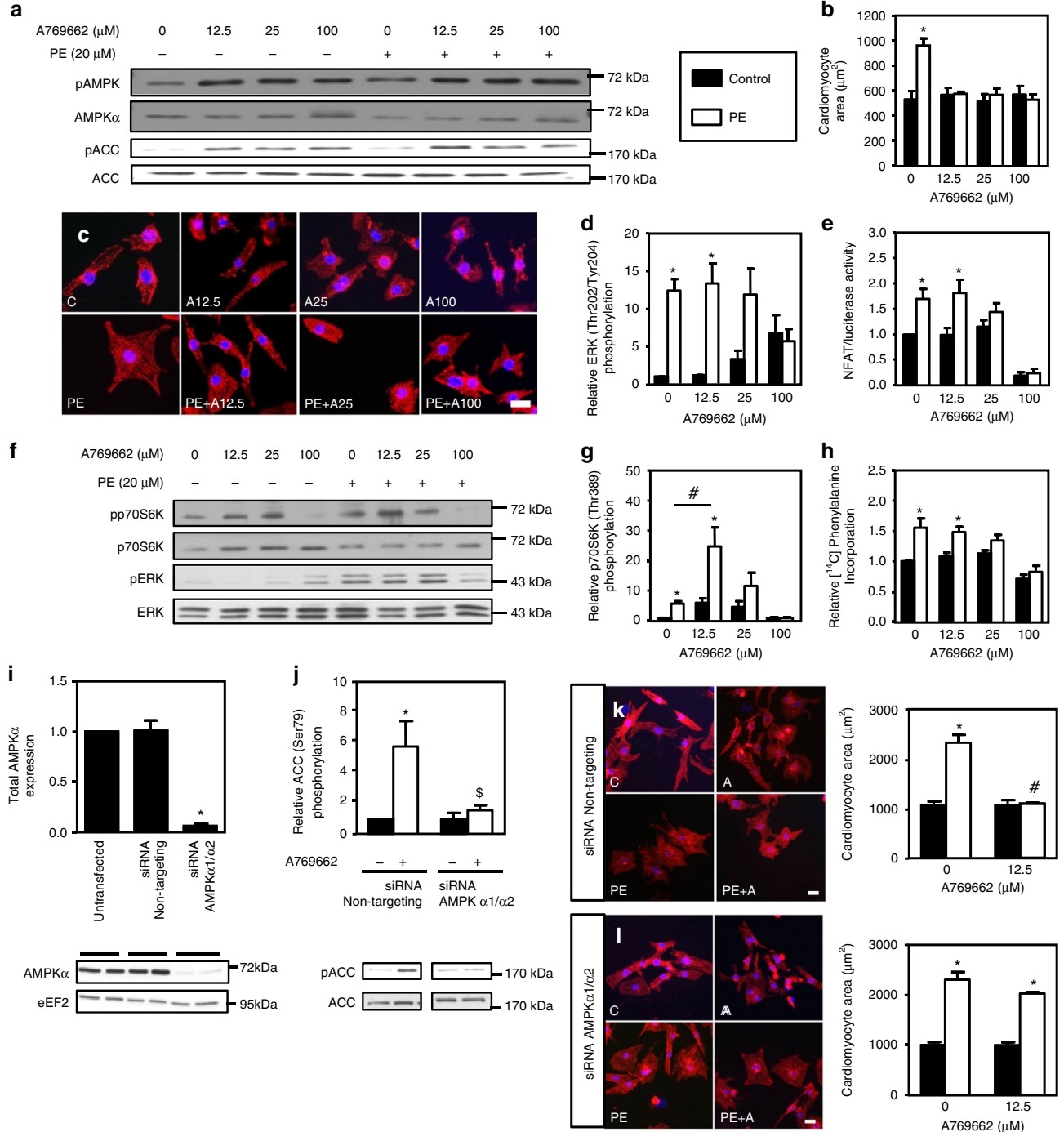

**Fig. 1** A769662 prevents NRVM hypertrophy. **a–h** NRVMs were treated with (open bars) or without (solid bars) phenylephrine (PE, 20 μM) in the presence or absence of increasing concentration of A769662 (from 12.5 to 100 μM) for 24 h except for ERK1/2 phosphorylation which has been evaluated after 1 h. **a** Representative immunoblot of AMPK$^{Thr172}$ and ACC$^{Ser79}$ phosphorylation. **b**, **c** Representative images and quantification of cardiomyocyte area evaluated after α-actinin immunostaining. Scale bar, 20 μm. $N = 3$. **d** Quantification of ERK$^{Thr202/Tyr204}$ phosphorylation. $N = 3$. **e** Evaluation of NFAT transcriptional activity by luciferase activity. $N = 3$. **f** Representative immunoblots of p70S6K$^{Thr389}$ and ERK$^{Thr202/Tyr204}$ phosphorylation. **g** Quantification of p70S6K$^{Thr389}$ phosphorylation. $N = 4$. **h** Amino acids incorporation into proteins measured by [$^{14}$C]-phenylalanine incorporation. $N = 3$. **i–l** NRVMs were transfected with control non-targeting siRNA or AMPKα1/α2 siRNA (50 nM) for 66 h. Then, NRVMs were treated with (open bars) or without (solid bars) phenylephrine (PE, 20 μM) in the presence or absence of A769662 (12.5 μM) for 24 h. **i** Representative immunoblot and quantification of total AMPKα. $N = 3$. **j** Representative immunoblot and quantification of ACC$^{Ser79}$ phosphorylation. $N = 3$. **k** Representative images and quantification of cardiomyocyte area evaluated after α-actinin immunostaining of NRVMs transfected with non-targeting siRNA. Scale bar, 20 μm. $N = 3$. **l** Representative images and quantification of cardiomyocyte area evaluated after α-actinin immunostaining of NRVMs transfected with AMPKα1/α2 siRNA. Scale bar, 20 μm. $N = 3$. Data in (**a–l**) are mean ± s.e.m. The data were analyzed using One-way ANOVA followed by Bonferroni post-test in (**i**) and Two-way ANOVA followed by Bonferroni post-test in (**b**, **d**, **e**, **g**, **h**, and **j–l**). *$p < 0.05$ vs. untreated cells, #$p < 0.05$ vs. corresponding PE-treated cells, $$p < 0.05$ vs. cells transfected with non-targeting siRNA. eEF2 was used as a loading control

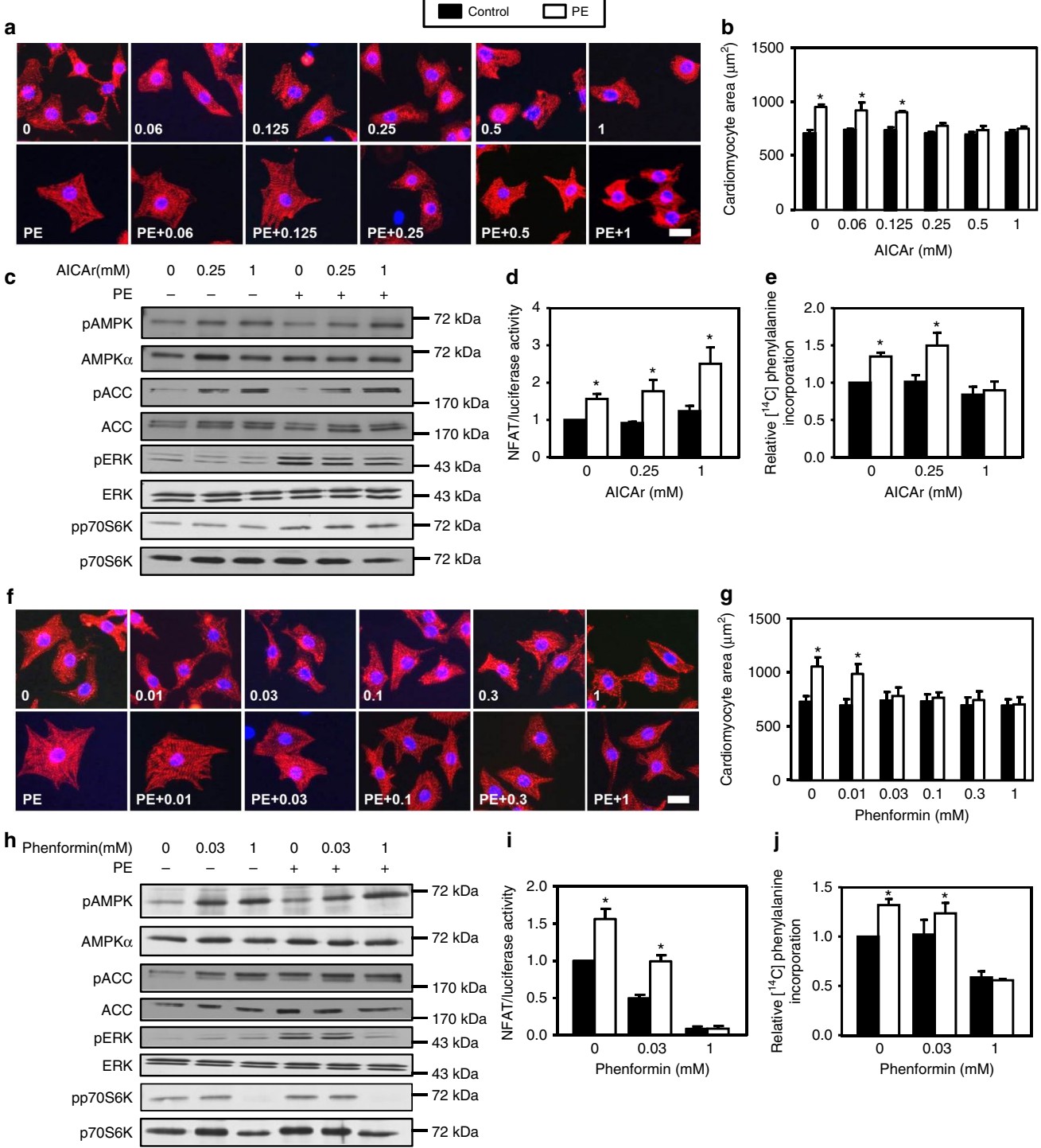

**Fig. 2** AICAr and phenformin mimick A769662 effects. **a–e** NRVMs were treated with (open bars) or without (solid bars) phenylephrine (PE, 20 μM) in the presence or not of AICAr (from 0.06 to 1 mM) for 24 h except for ERK1/2 phosphorylation which has been evaluated after 1 h. **a, b** Representative images and quantification of cardiomyocyte area evaluated after α-actinin immunostaining. Scale bar, 20 μm. $N = 3$. **c** Representative immunoblots of AMPK[Thr172], ACC[Ser79], ERK[Thr202/Tyr204] and p70S6K[Thr389] phosphorylation. **d** Evaluation of NFAT transcriptional activity by luciferase activity. $N = 3$. **e** Amino acids incorporation into proteins measured by [14C]-phenylalanine incorporation. $N = 5$. **f–j** NRVMs were treated with (open bars) or without (solid bars) phenylephrine (PE, 20 μM) in the presence or not of phenformin (from 0.01 to 1 mM) for 24 h except for ERK1/2 phosphorylation which has been evaluated after 1 h. **f, g** Representative images and quantification of cardiomyocyte area evaluated after α-actinin immunostaining. Scale bar, 20 μm. $N = 3$. **h** Representative immunoblots of AMPK[Thr172], ACC[Ser79], ERK[Thr202/Tyr204], and p70S6K[Thr389] phosphorylation. **i** Evaluation of NFAT transcriptional activity by luciferase activity. $N = 3$. **j** Amino acids incorporation into proteins measured by [14C]-phenylalanine incorporation. $N = 4$. Data in (**a–j**) are mean ± s.e.m. The data in (**b, d, e, g,** and **i, j**) were analyzed using Two-way ANOVA followed by Bonferroni post-test. *$p < 0.05$ vs. untreated cells

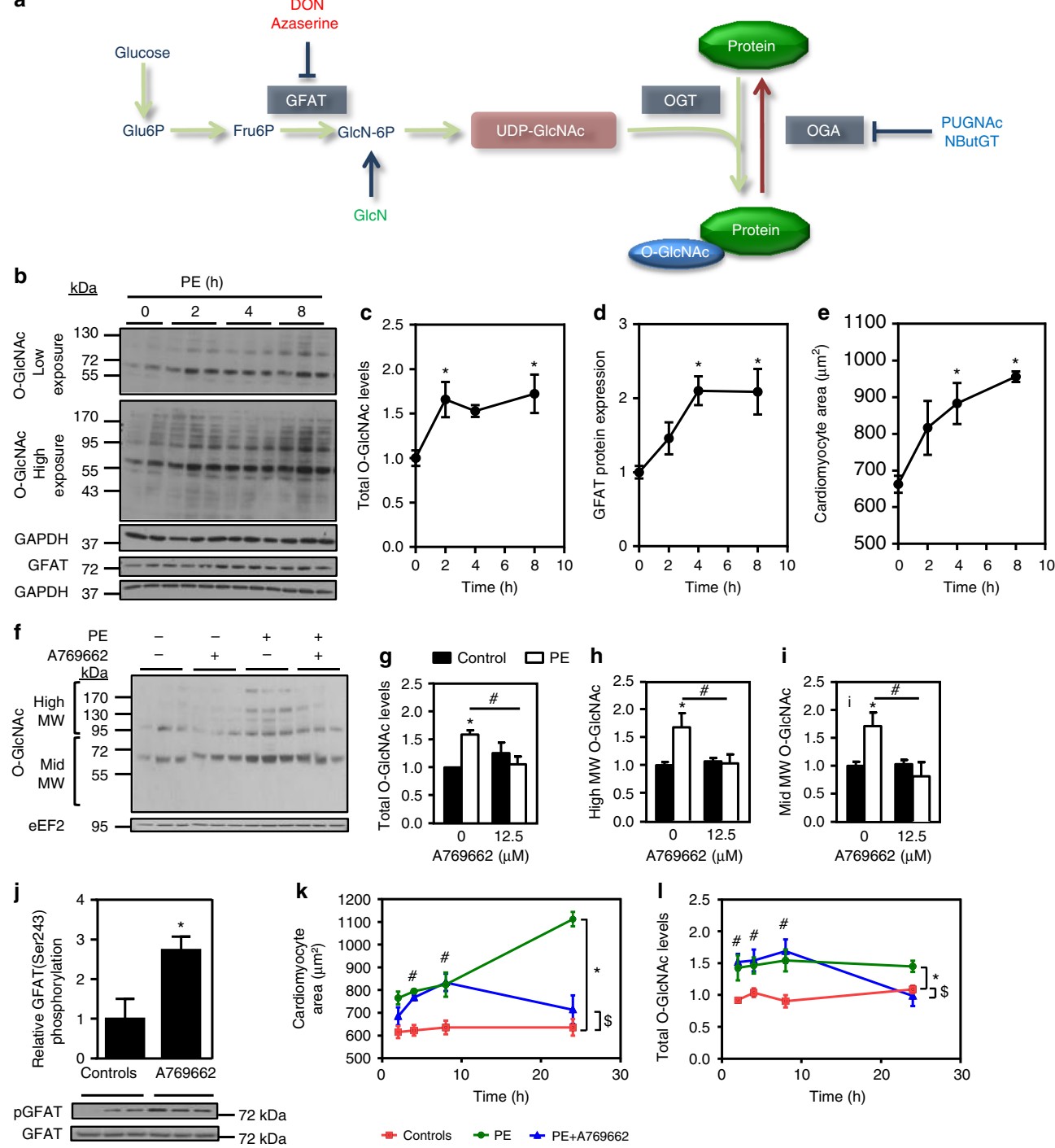

**Fig. 3** AMPK activation reduces protein O-GlcNAcylation. **a** Schematic representation of the HBP/O-GlcNAcylation pathway. **b–e** NRVMs were treated with phenylephrine (PE, 20 μM) for increasing time periods (from 2 to 8 h). **b** Representative immunoblots of protein O-GlcNAcylation levels and GFAT protein expression. **c** Quantification of protein O-GlcNAcylation levels. $N = 5$. **d** Quantification of GFAT protein expression. $N = 5$. **e** Quantification of cardiomyocyte area evaluated after α-actinin immunostaining. $N = 3$. **f–i** NRVMs were treated with (open bars) or without (solid bars) phenylephrine (PE, 20 μM) in the presence or not of A769662 (12.5 μM) for 24 h. Representative immunoblot and quantification of protein O-GlcNAcylation levels. $N = 3$. **j** Representative immunoblot and quantification of GFAT$^{Ser243}$ phosphorylation in NRVMs treated 1 h with A769662. $N = 3$. **k–l** NRVMs were treated with phenylephrine (PE, 20 μM) in the presence or absence of A769662 (12.5 μM) for increasing time periods (from 2 to 24 h). **k** Quantification of cardiomyocyte area evaluated after α-actinin immunostaining. $N = 3$–6. **l** Quantification of protein O-GlcNAcylation levels. $N = 5$–7. Data in (**c–l**) are mean ± s.e.m. The data were analyzed using Two-way ANOVA followed by Bonferroni post-test in (**g–i**) and (**k**, **l**), One-way Anova followed by Bonferroni post-test in (**c–e**) and unpaired Student's *t*-test in (**j**). *$p < 0.05$ vs. untreated cells, #$p < 0.05$ vs. PE-treated cells for (**c–e** and **j**. *$p < 0.05$ global effect of PE vs. controls, $$p < 0.05$ global effect of PE + A vs. controls and #$p < 0.05$ PE + A vs. controls at each time point for (**k**, **l**). GAPDH and eEF2 were used as loading control, MW molecular weight

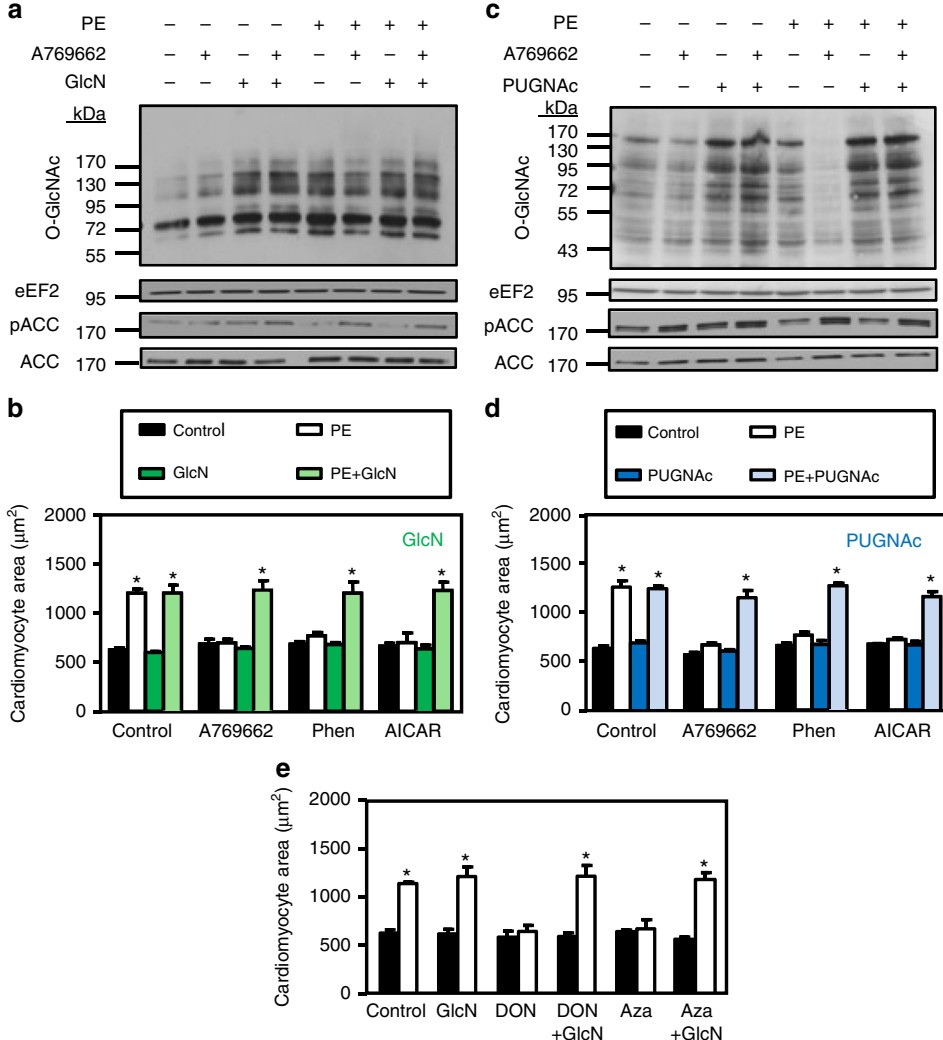

**Fig. 4** Glucosamine or PUGNAc prevents the anti-hypertrophic action of AMPK. **a**–**d** NRVMs were treated with (open bars) or without (solid bars) phenylephrine (PE, 20 µM) in the presence or absence of A769662 (12.5 µM), phenformin (phen, 0.03 mM), AICAr (0.25 mM), PUGNAc (50 µM) and/or glucosamine (GlcN, 5 mM) for 24 h. **a** Representative immunoblot of protein O-GlcNAcylation levels and ACC[Ser79] phosphorylation in GlcN experiments. **b** Effect of GlcN on the anti-hypertrophic action of AMPK activators. $N = 3$. **c** Representative immunoblot of protein O-GlcNAcylation levels and ACC[Ser79] phosphorylation in PUGNAc experiments. **d** Effect of PUGNAc on the anti-hypertrophic action of AMPK activators. $N = 3$. **e** Quantification of cardiomyocyte area of NRVMs treated with (open bars) or without (solid bars) phenylephrine (PE, 20 µM) in the presence or not of Azaserine (Aza, 5 µM), DON (20 µM) and/or glucosamine (GlcN, 5 mM) for 24 h. $N = 3$. Data in (**b**, **d** and **e**) are mean ± s.e.m. The data were analyzed using Two-way ANOVA followed by Bonferroni post-test in (**b**, **d** and **e**). *$p < 0.05$ vs. untreated cells. eEF2 was used as loading control, MW molecular weight

NRVMs were maintained and tested in the presence of high glucose (25 mM). Because O-GlcNAc levels are known to be dependent of extracellular glucose concentration[36], the same experiments were conducted on NRVMs incubated in the presence of 5 mM glucose (Supplementary Fig. 5a, b). The anti-hypertrophic action of A769662 was equally abolished by PUGNAc and GlcN, ruling out any unspecific effects due to particular glucose availability in medium.

Similarly, we evaluated the effect of A769662 on hypertrophied adult rat ventricular myocytes (ARVMs) in the presence of 5 mM glucose (Supplementary Fig. 6a-d). As described previously[37], treatment of ARVMs with 100 µM PE induced 15% hypertrophy and this was totally blunted by low-dose of A769662 (Supplementary Fig. 6a, b). This was associated with a decrease in O-GlcNAcylation, and PUGNAc reversed A769662's effects on O-GlcNAc levels and ARVM size (Supplementary Fig. 6a-d).

Taken altogether, these results in cultured cardiomyocytes provide strong evidence that AMPK inhibits hypertrophy by targeting GFAT and lowering O-GlcNAcylation induced by pro-hypertrophic agents. Conversely, increased HBP/O-GlcNAc signaling is sufficient to counteract the anti-hypertrophic action of AMPK activators.

**O-GlcNAc inhibition by AMPK blocks hypertrophy in vivo.** Our hypothesis was also tested in vivo. First, we took advantage of mice deficient in AMPKα2 (AMPKα2 KO), the main isoform of the AMPK catalytic subunit expressed in the heart, to evaluate the impact of AMPK deletion on basal cardiac O-GlcNAc levels (Fig. 5a-e and Supplementary Fig. 7). Even though AMPKα2 KO mice have been shown to possess a normal basal cardiac phenotype[29, 38], O-GlcNAc levels were significantly elevated in the hearts of these mice compared to their wild type (WT) control littermates (Fig. 5a, b and Supplementary Fig. 7) and this was associated with a decreased in OGA protein level whereas GFAT and OGT expression was unchanged (Fig. 5c–e). Similar to the

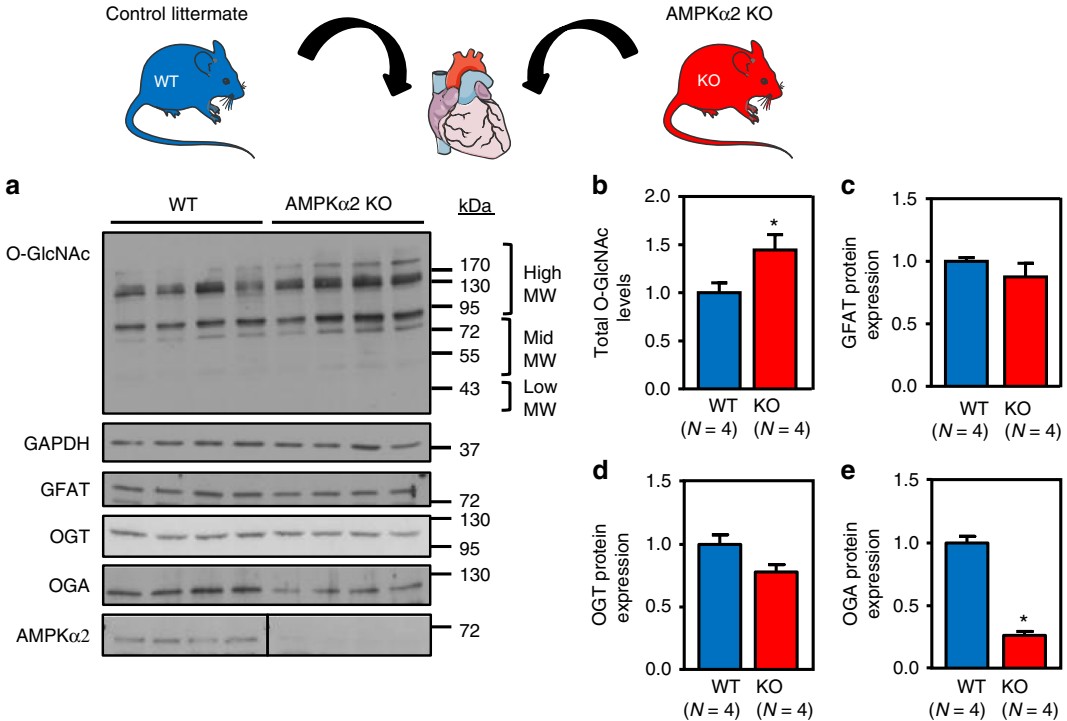

**Fig. 5** O-GlcNAc levels are increased in AMPKα2 KO mice. **a–e** Representative immunoblot and quantification of protein O-GlcNAcylation levels (**b**), GFAT (**c**), OGT (**d**), OGA (**e**) and AMPKα2 expression in AMPKα2 KO mice compared to control littermates (WT). $N = 4$. Data in **b–e** are expressed as mean ± s. e.m. The data were analyzed using unpaired Student's $t$-test. $*p < 0.05$ vs. WT. GAPDH was used as loading control, MW molecular weight

data obtained in cultured cells with GlcN or PUGNAc alone, these data reveal that an increase in basal O-GlcNAc levels induced by AMPK deletion are not sufficient to induce cardiac hypertrophy per se.

Next, we subjected WT and AMPKα2 KO mice to continuous infusion of AngII, a pro-hypertrophic treatment frequently used[39] (Fig. 6a). In agreement with data obtained with other pro-hypertrophic treatments[29, 38], AngII promoted left ventricular (LV) hypertrophy in both mouse strains (Supplementary Table 1). More importantly, cardiac O-GlcNAc levels were significantly augmented in WT animals after 5 days of AngII treatment compared to the untreated controls (Fig. 6b, c and Supplementary Fig. 7). These results are in agreement with our in vitro data obtained in NRVMs/ARVMs and concur similarly to what has been reported by Lunde and colleagues in other models of cardiac hypertrophy[22]. Conversely, O-GlcNAc levels in AMPKα2 KO hearts, which were already elevated under basal conditions, were unchanged upon AngII treatment (Fig. 6b, c and Supplementary Fig. 7), suggesting that AMPK deletion promotes global O-GlcNAc modifications that are very similar to those induced by pro-hypertrophic agents.

To evaluate the impact of AMPK activation on those phenomena, both WT and KO mice, treated or not with AngII, were co-treated with metformin. In contrast to A769662, characterized by poor oral bioavailability[24], metformin is a more efficient cardiac AMPK activator in vivo[40, 41]. As expected, metformin efficiently increased AMPK signaling (followed by ACC phosphorylation) in WT but not in AMPKα2 KO hearts (Supplementary Fig. 8). It also inhibited the AngII-induced increase in LV mass and cardiomyocyte size of WT animals (Fig. 7a–c and Supplementary Table 1) without any significant effect on systolic blood pressure (129 ± 2, 149 ± 10 and 162 ± 8 mmHg for control, AngII and AngII + metformin, respectively). Absence of those anti-hypertrophic effects in AMPKα2 KO hearts

demonstrate an AMPK dependency of the metformin action (Fig. 7d-f and Supplementary Table 1).

The next step was to confirm the role of O-GlcNAcylation in the anti-hypertrophic action of AMPK in vivo (Fig. 8a–g). In line with in vitro data, the anti-hypertrophic effect of metformin in WT mice was associated with a reduction of cardiac O-GlcNAc levels (Fig. 8d, e and Supplementary Fig. 7). In contrast, O-GlcNAc levels were not affected by metformin treatment in hearts from AMPKα2 KO animals confirming the role of AMPK in metformin effect on O-GlcNAcylation regulation (Fig. 8f, g and Supplementary Fig. 7). To further substantiate that AMPK primarily regulates cardiac hypertrophy by modulating O-GlcNAcylation, a group of WT mice were co-treated with 1,2-dideoxy-2′-propyl-alpha-d-glucopyranoso-[2,1-D]-Delta 2′-thiazoline (NButGT), an OGA inhibitor compatible with chronic utilization in vivo. We first confirmed that intraperitoneal injection of this compound promoted cardiac protein O-GlcNAcylation (Supplementary Fig. 9). Like for GlcN or PUGNAc in vitro, NButGT alone did not induce cardiac hypertrophy development (Fig. 8a–c) providing robust evidence that an increase in global O-GlcNAcylation is not sufficient to promote cardiac hypertrophy without pro-hypertrophic co-treatment. More importantly, NButGT reduced the impact of metformin on protein O-GlcNAcylation levels (Fig. 8d, e and Supplementary Fig. 7) and prevented its anti-hypertrophic effect in the presence of AngII (Fig. 8a–c), confirming the central role of O-GlcNAcylation in the anti-hypertrophic action of AMPK.

Similar studies were conducted in a model of cardiac hypertrophy induced after TAC surgery. This TAC-induced model of cardiac hypertrophy has been previously characterized in terms of LV mass and function[42]. Three weeks of TAC induced a significant increase in cardiomyocyte size in WT mice (Supplementary Fig. 10a, b). Like for AngII studies, metformin significantly reduced this TAC-mediated increase in

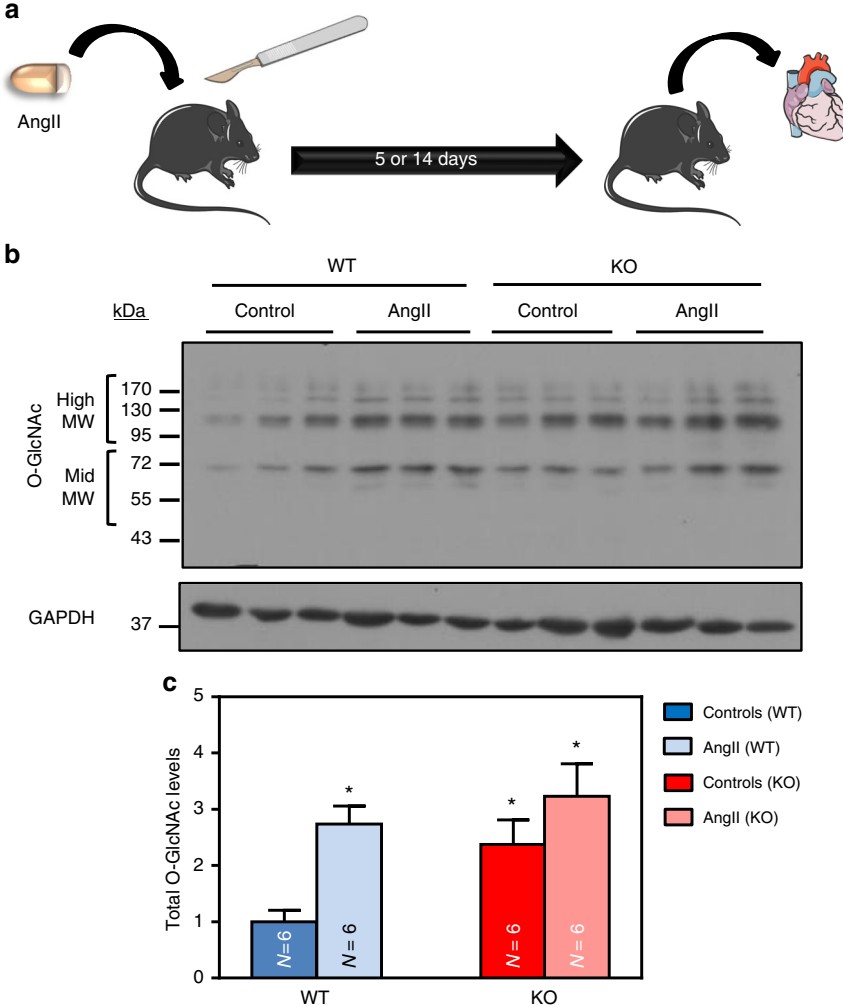

**Fig. 6** AngII treatment increases O-GlcNAc levels in the presence of AMPK. WT (in blue) and AMPKα2 KO (in red) mice were treated with or without angiotensin II (AngII, 2 mg/kg/d) for 5 days in order to evaluate O-GlcNAcylated protein levels. **a** Schematic representation of the experimental protocol where mice were treated for 5 or 14 days in order to evaluate O-GlcNAcylated protein levels or cardiac hypertrophy respectively. **b** Representative immunoblot of protein O-GlcNAcylation levels in WT and AMPKα2 KO mouse hearts. **c** Quantification of protein O-GlcNAcylation levels in WT and AMPKα2 KO mouse hearts. $N = 6$. Data in **c** are expressed as mean ± s.e.m and were analyzed using Two-way ANOVA followed by Bonferroni post-test. *$p < 0.05$ vs. untreated WT mice. GAPDH is used as loading control, MW molecular weight

cardiomyocyte size in WT mice and NButGT counteracted metformin action (Supplementary Fig. 10a, b). In contrast, cardiomyocyte size was not affected by metformin in KO mice (Supplementary Fig. 10c, d). So, the interplay between AMPK and O-GlcNAc signaling in cardiac hypertrophy prevention could be seen not only in hormonal model of cardiac hypertrophy, but also in pressure induced-cardiac hypertrophy.

**AMPK decreases O-GlcNAcylation in vivo via GFAT and OGT.** Similarly to what we observed in vitro, it has already been shown that O-GlcNAcylation occurring during cardiac hypertrophy correlated with an increase in GFAT protein level[22, 43]. We confirm these data in our AngII model (Fig. 9a). As observed in vitro using A769662, metformin did not reverse this increase in GFAT expression (Fig. 9a) but promoted its phosphorylation on its inhibitory Ser243 site in an AMPK-dependent manner (Fig. 9b–e), as this increase in phosphorylation disappeared in AMPKα2 KO mouse hearts (Fig. 9f–j). Even if GFAT regulation appears as a critical piece of the puzzle in regulation of O-GlcNAcylation by AMPK, our in vivo study highlighted another

enzyme involved in the interaction between HBP and AMPK. Indeed, AngII-mediated increase in O-GlcNAcylation was also associated with elevation of OGT protein level in WT but not AMPKα2 KO hearts (Fig. 10a). Metformin reduced this increase in OGT protein level in WT mice (Fig. 10c) but not in AMPKα2 KO hearts (Fig. 10b, d). By contrast, OGA expression was unaltered by AngII and metformin (Supplementary Fig. 11).

**AMPK activation prevents the O-GlcNAcylation of troponin T.** The exact role of O-GlcNAcylation in cardiac hypertrophy development is still largely unclear. However, growing evidences in the literature show that important players of cardiac hypertrophy are potentially O-GlcNAcylable[18]. For this reason, we evaluated the O-GlcNAcylation level of troponin T (TnT) for which increased O-GlcNAcylation was associated with development of adverse cardiac remodeling[44]. Interestingly, O-GlcNAcylation level of TnT was increased by AngII whereas metformin treatment prevented this increase in WT mice (Fig. 10e). This metformin-mediated inhibition of TnT O-GlcNAcylation disappeared in AMPKα2 KO mouse hearts

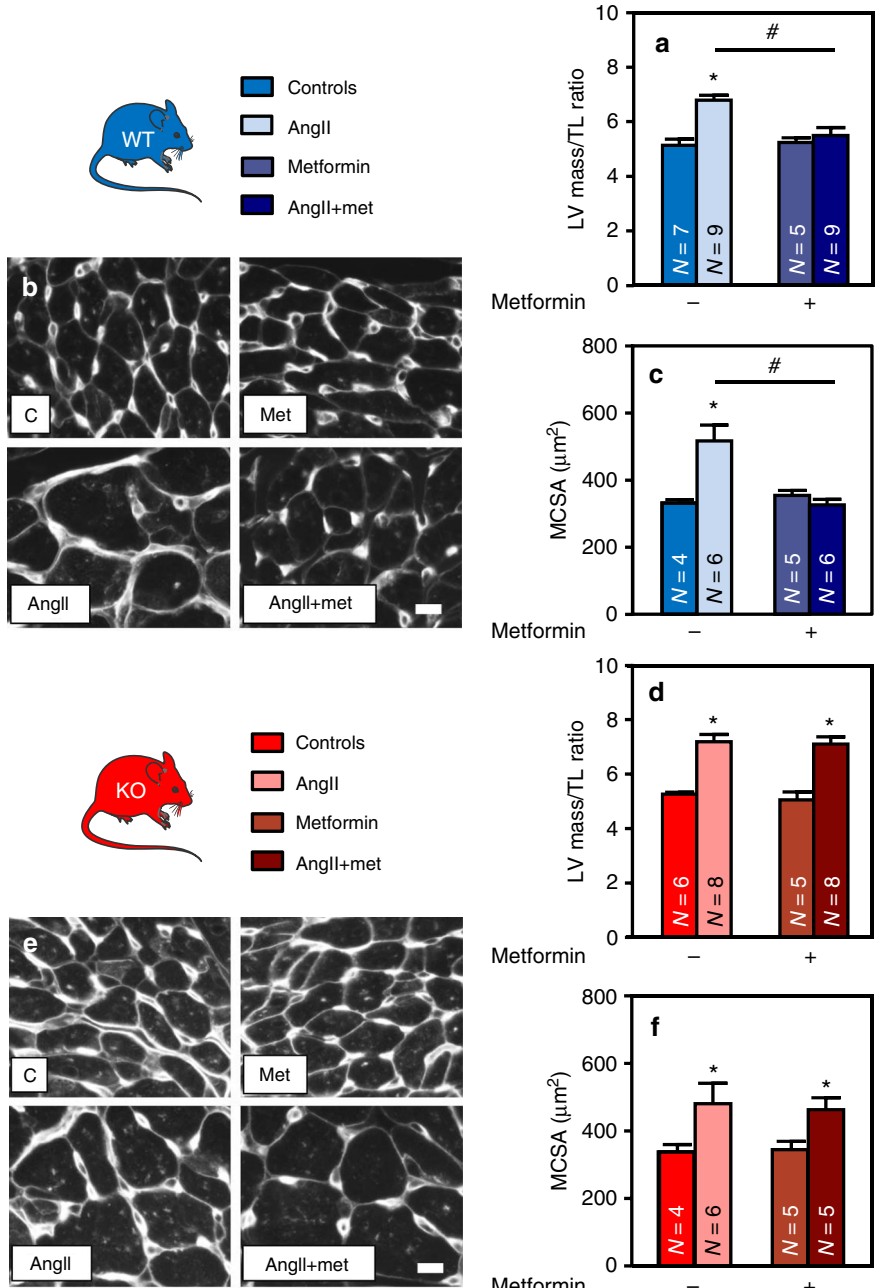

**Fig. 7** Metformin prevents cardiac hypertrophy in an AMPK-dependent manner. **a–f** WT (in blue) and AMPKα2 KO (in red) mice were treated with or without angiotensin II (AngII, 2 mg/kg/d) in the presence or absence of metformin (met, 200 mg/kg/d) for 14 days in order to evaluate cardiac hypertrophy. **a** Left ventricular (LV) mass on tibia length (TL) ratio in WT mice. $N = 5–9$ **b–c** Representative images and quantification of myocyte cross sectional area evaluated after WGA staining in WT mouse hearts. Scale bar, 10 μm. $N = 4–6$ **d** LV mass on TL ratio in AMPKα2 KO mice. $N = 5–8$ **e–f** Representative images and quantification of myocyte cross sectional area evaluated after WGA staining in AMPKα2 KO mouse hearts. Scale bar, 10 μm. $N = 4–6$. Data in (**a**, **c**, **d**, and **f**) are expressed as mean ± s.e.m. and were analyzed using Two-way ANOVA followed by Bonferroni post-test. *$p < 0.05$ vs. untreated mice, #$p < 0.05$ vs. AngII-treated mice

(Fig. 10f), proving that TnT is one of the indirect downstream targets of AMPK in this process.

## Discussion

The major findings of this study are that: (i) AMPK activation by A769662 blocks cardiac hypertrophy development; (ii) three different AMPK activators, used at low concentrations, consistently inhibit cardiomyocyte hypertrophy without detectable changes of downstream targets previously suggested to participate in this phenomenon; (iii) increase in protein O-GlcNAcylation is required for cardiomyocyte hypertrophy in vitro; (iv) inhibition of cardiac hypertrophy by AMPK is tightly associated with reduction of increased O-GlcNAcylation in vitro as well as in vivo; (v) such association disappeared in AMPK-deficient mouse model; (vi) counteracting inhibition of O-GlcNAcylation abolished AMPK's anti-hypertrophic action in WT cellular and animal models of cardiac hypertrophy. Altogether, our results establish a new concept in which AMPK blocks cardiac hypertrophy by predominantly modulating O-GlcNAcylation (summarized in Fig. 10g).

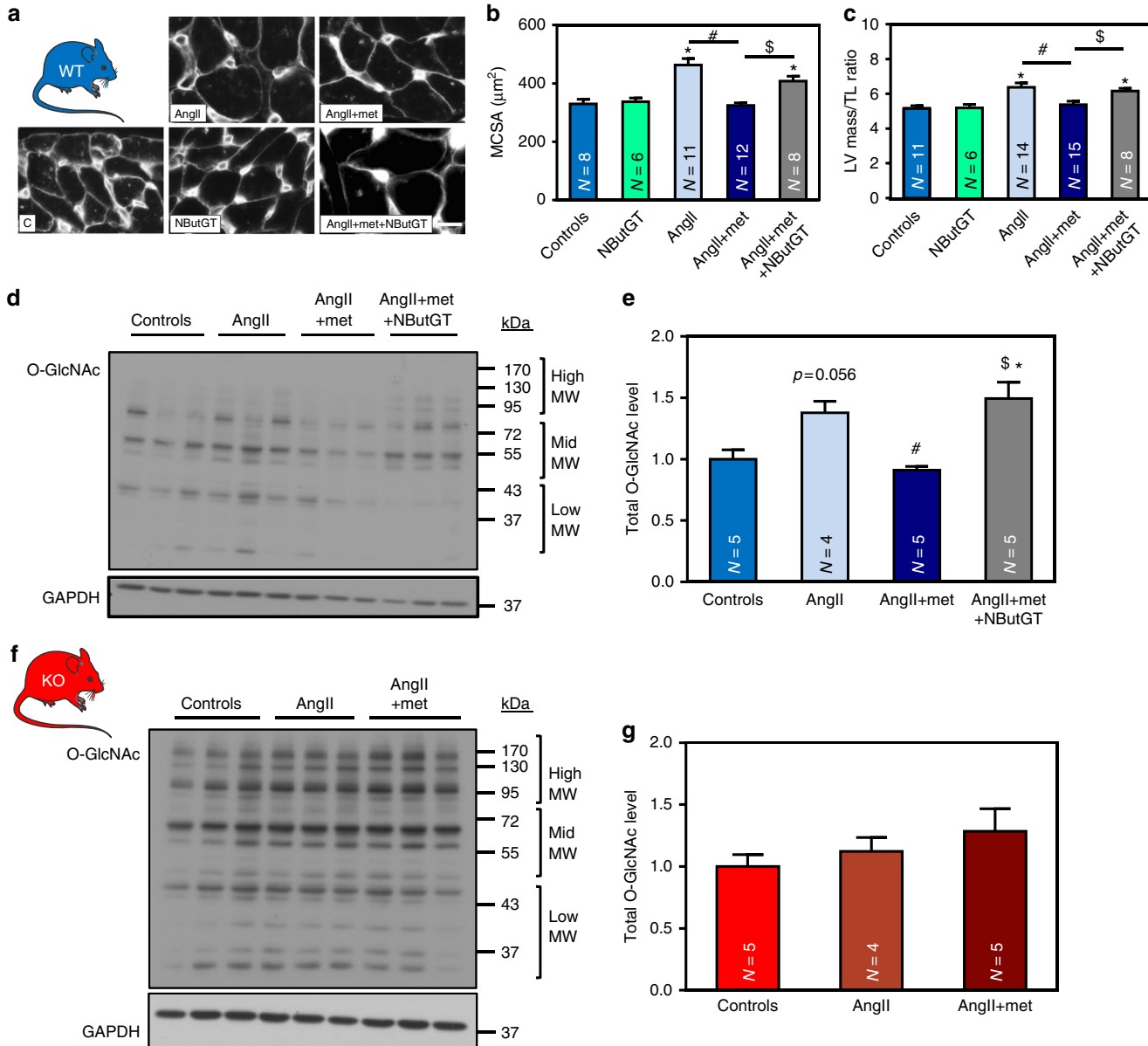

**Fig. 8** NButGT prevents the anti-hypertrophic effect of metformin. **a–g** WT (in blue) and AMPKα2 KO (in red) mice were treated with or without angiotensin II (AngII, 2 mg/kg/d) in the presence or absence of metformin (met, 200 mg/kg/d). Two groups of WT mice also received NButGT (50 mg/kg/d). **a**, **b** Representative images and quantification of myocyte cross sectional area after WGA staining in WT mouse heart. Scale bar, 10 μm. $N = 6$–12 (**c**) Left ventricular (LV) mass on tibia length (TL) ratio in WT mice. $N = 6$–15 **d**, **e** Representative immunoblot and quantification of total O-GlcNAc levels in WT mouse hearts. $N = 4$–5 **f**, **g** Representative immunoblot and quantification of total O-GlcNAc levels in AMPKα2 KO mouse hearts. $N = 4$–5. Data in (**b**, **c**, **e**, and **g**) are expressed as mean ± s.e.m. and were analyzed using One-way ANOVA followed by Bonferroni post-test. *$p < 0.05$ vs. untreated mice, #$p < 0.05$ vs. AngII-treated mice, $$p < 0.05$ vs. AngII + met-treated mice. GAPDH was used as loading control, MW molecular weight

Pharmacological agents previously known to prevent PE-induced cardiomyocyte hypertrophy, such as AICAr or the biguanide metformin[11, 12], act as AMP analogs or increase AMP concentration through inhibition of mitochondrial respiration, and, thus, are not specific to AMPK[45]. Our study was initially conducted to investigate the anti-hypertrophic action of the more specific AMPK activator A769662. Here, we show that this compound successfully blocks cardiomyocyte hypertrophy induced by PE or AngII. It is interesting to note that A769662 does not equally bind all AMPK heterotrimers but rather preferentially activates cardiac AMPK heterotrimeric complexes containing β1 subunit[46]. In other words, the activation of these β1-containing AMPK heterotrimers is probably sufficient to block hypertrophy. This should be considered in conjunction with our

in vivo data but also previous studies[13, 15], demonstrating that deletion of the AMPKα2 catalytic isoform is sufficient to prevent hypertrophy inhibition induced by AMPK activators. Inasmuch as we also recently reported that AMPKα2 binds preferentially to AMPKβ1 in cardiomyocytes[26], this makes AMPKα2/β1 pivotal target to reduce cardiac hypertrophy.

More importantly, our dose-response studies using A769662, but also phenformin and AICAr, allowed us to establish a novel concept (Fig. 10g). We showed that these three AMPK activators prevent NRVM hypertrophy without affecting previously-proposed AMPK targets, including p70S6K, NFAT, and ERK. We conclude from these in vitro data that these major pro-hypertrophic pathways are required but not sufficient to induce cardiomyocyte hypertrophy. In other words, other pro-

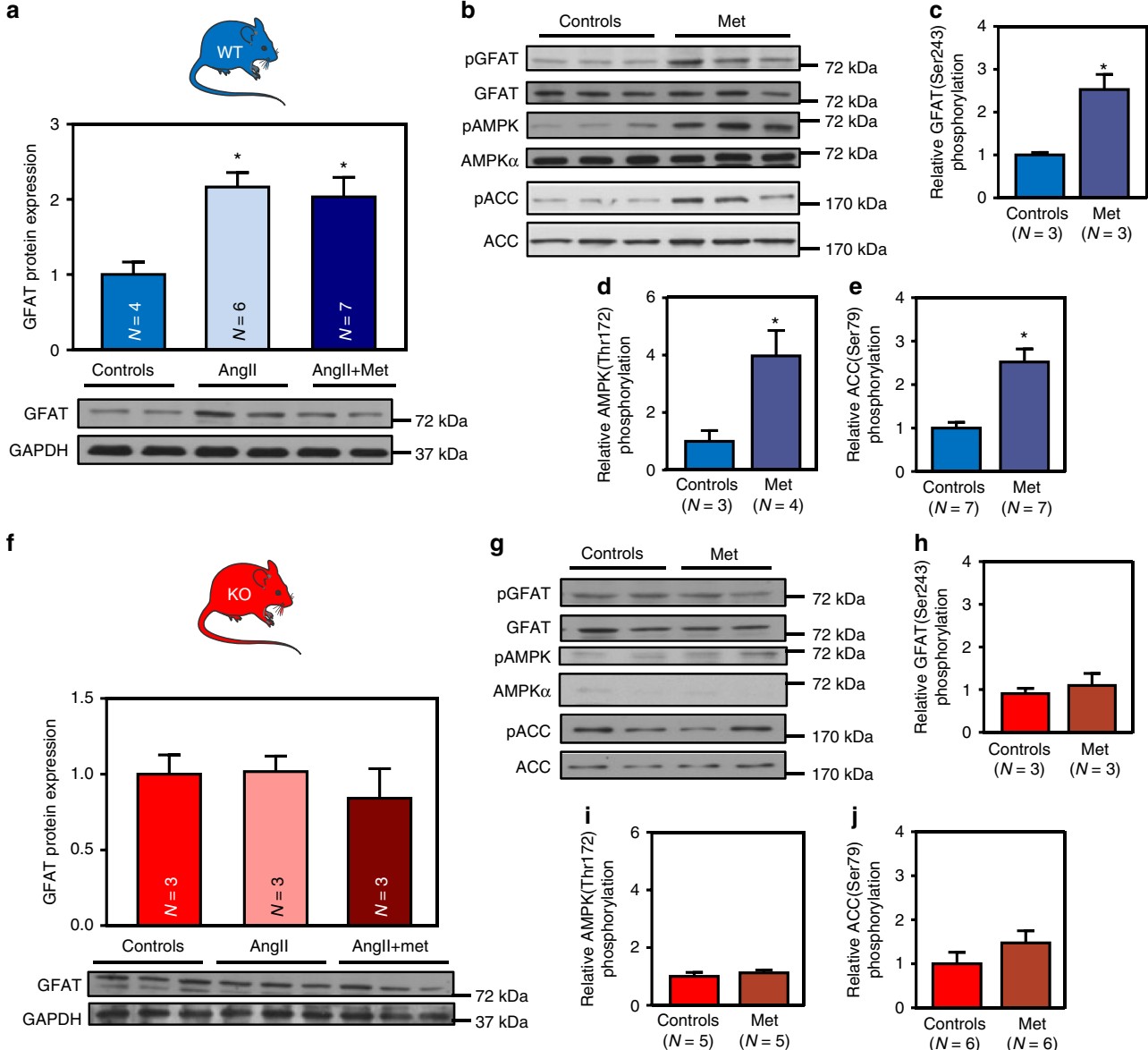

**Fig. 9** AMPK activation promotes GFAT phosphorylation in vivo. **a–j** WT (in blue) and AMPKα2 KO (in red) mice were treated with or without angiotensin II (AngII, 2 mg/kg/d) in the presence or absence of metformin (met, 200 mg/kg/d). **a** Representative immunoblot and quantification of GFAT protein expression in WT mouse hearts. $N = 4$–7 **b** Representative immunoblots of GFAT[Ser243], AMPK[Thr172], and ACC[Ser79] phosphorylation in WT mouse hearts. **c** Quantification of GFAT[Ser243] phosphorylation in WT mouse hearts. $N = 3$ **d** Quantification of AMPK[Thr172] phosphorylation in WT mouse hearts. $N = 3$–4 **e** Quantification of ACC[Ser79] phosphorylation in WT mouse hearts. $N = 7$ **f** Representative immunoblot and quantification of GFAT protein expression in AMPKα2 KO mouse hearts. $N = 3$ **g** Representative immunoblots of GFAT[Ser243], AMPK[Thr172], and ACC[Ser79] phosphorylation in AMPKα2 KO mouse hearts. **h** Quantification of GFAT[Ser243] phosphorylation in AMPKα2 KO mouse hearts. $N = 3$ **i** Quantification of AMPK[Thr172] phosphorylation in AMPKα2 KO mouse hearts. $N = 5$ **j** Quantification of ACC[Ser79] phosphorylation in AMPKα2 KO mouse hearts. $N = 6$. Data in (**a–j**) are expressed as mean ± s.e.m. The data were analyzed using One-way ANOVA followed by Bonferroni post-test in (**a**, **f**) and unpaired Student's $t$-test in (**c–e** and **h–j**). *$p < 0.05$ vs. untreated mice. GAPDH was used as loading control

hypertrophic mechanisms co-exist, are equally important and should be targeted by AMPK activators at low concentration. In our quest to find this missing link, our attention was drawn to O-GlcNAcylation process. O-GlcNAcylation was first mentioned in the context of cardiac hypertrophy by Young and colleagues[43] and this has been confirmed by several other groups[22, 23, 47, 48]. However, its exact role remains largely unclear. Interestingly, several important players of cardiac hypertrophy including proteins involved in autophagy, in atrogenic pathway, in calcium handling, in endoplasmic/sarcoplasmic reticulum stress and in contractile machinery have been discovered to be potentially O-

GlcNAcylated in the heart under particular pathophysiological contexts[18, 49, 50]. Here, we demonstrate that protein O-GlcNAcylation is promoted by pro-hypertrophic treatments in both NRVMs and ARVMs as well as in the heart in vivo. Moreover, it appears from our data that O-GlcNAcylation elevation is not just a marker of cardiac hypertrophy but could also be considered as a prerequisite for this pathological process. First, O-GlcNAcylation occurs early in the hypertrophic process (within 2 h of pro-hypertrophic treatment in our in vitro NRVM model and within the first 5 days of in vivo treatment in AngII-treated mice). Second, O-GlcNAc levels perfectly correlate to

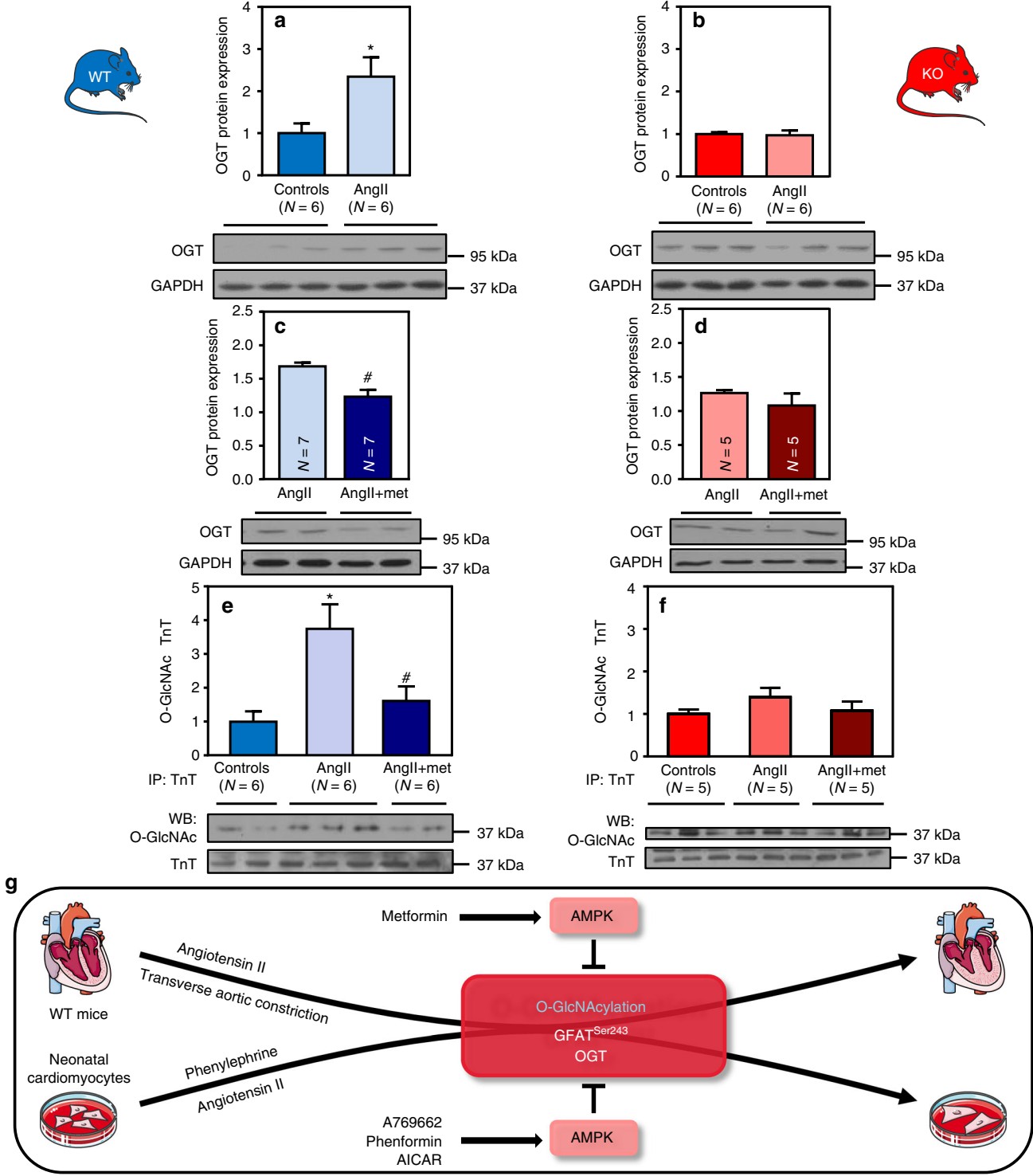

**Fig. 10** AMPK activation regulates OGT expression and troponin T O-GlcNAc level. **a**, **b** WT (in blue) and AMPKα2 KO (in red) mice were treated with or without Angiotensin II (AngII, 2 mg/kg/d) for 5 days. **a** Representative immunoblot and quantification of OGT protein expression in WT mouse hearts. $N = 6$ **b** Representative immunoblot and quantification of OGT protein expression in AMPKα2 KO mouse hearts. $N = 6$ **c**, **d** WT (in blue) and AMPKα2 KO (in red) mice were treated with or without Angiotensin II (AngII, 2 mg/kg/d) in the presence or absence of metformin (met, 200 mg/kg/d) for 5 days. **c** Representative immunoblot and quantification of OGT protein expression in WT mouse hearts. $N = 7$ **d** Representative immunoblot and quantification of OGT protein expression in AMPKα2 KO mouse hearts. $N = 5$ **e** Representative immunoblot and quantification of O-GlcNAcylated troponin T (TnT) in WT mouse hearts. $N = 6$ **f** Representative immunoblot and quantification of O-GlcNAcylated TnT in AMPKα2 KO mouse hearts. $N = 5$ **g** Schematic model of the proposed interplay in the regulation of cardiac hypertrophy by O-GlcNAcylation and AMPK. Data in (**a**–**f**) are expressed as mean ± s.e.m. The data were analyzed using One-way ANOVA followed by Bonferroni post-test in (**e**, **f**) and unpaired Student's $t$-test in (**a**–**d**). *$p < 0.05$ vs. untreated mice, #$p < 0.05$ vs. AngII-treated mice. GAPDH was used as a loading control

cardiomyocyte size in all our experiments. Third, preventing this increase in O-GlcNAcylation by inhibiting GFAT via the use of DON or Aza is sufficient to block NRVM hypertrophy. In agreement, it has been shown that GFAT inhibition by DON prevents PE-induced hypertrophic markers to progress in NRVMs[23]. In contrast to these in vitro data, it has to be mentioned that a very recent study performed by the same research group and using inducible/cardio-specific OGT-deficient mice submitted to TAC surgery revealed that OGT does not appear to be essential for cardiac hypertrophy development[20]. However, the inducible animal model used in this study, the classical α-MHC-driven mutated estrogen receptor-flanked Cre recombinase, allowed only partial OGT deletion[20]. It is tempting to hypothesize that such partial deletion is not sufficient to have a significant impact on hypertrophy development. Nevertheless, further studies will be necessary to reach a definitive conclusion regarding this issue.

Even if required, at least in vitro, O-GlcNAcylation is not sufficient to promote hypertrophy per se. Indeed, GlcN, PUGNAc (in vitro) and NButGT (in vivo) are unable to induce hypertrophy in the absence of pro-hypertrophic agents. The same applies for AMPKα2 KO mice, which are characterized by increased levels of O-GlcNAc under basal state without any significant effect on heart size under basal conditions. This highly suggests that O-GlcNAcylation must be combined with other molecular events, such as the increase in protein synthesis and modifications in gene transcription to promote cardiac hypertrophy.

Beyond and above all other consideration, the most significant findings of our study reside in two other observations. First, AMPK activation (regardless of the activator used) reverses the increase in protein O-GlcNAcylation occurring during cardiac hypertrophy development in all in vitro and in vivo models investigated so far. Second, counteracting this AMPK-mediated inhibition of O-GlcNAcylation with GlcN and/or OGA inhibitors (PUGNAc and NButGT) totally neutralizes the anti-hypertrophic effect of this protein kinase in vitro as well as in vivo. As far as we know, this is the first proof of a crucial role of O-GlcNAcylation downstream of AMPK in its action against cardiac hypertrophy development. Our study is supported by evidence of crosstalk between O-GlcNAc and AMPK in different cell types, including myotubes, adipocytes, hepatocytes, neuronal, and endothelial cells[51–56].

Insofar as maximal AMPK activation inhibits not only O-GlcNAcylation but also other downstream targets (e.g., p70S6K, NFAT, ERK), one may wonder regarding the relative importance of O-GlcNAcylation inhibition in the overall anti-hypertrophic action of AMPK. Results from our in vivo study help to answer this question. Indeed, in our work, mice were treated with a relatively high dose of metformin. Such dose has been previously shown to inhibit classical AMPK downstream targets, including pathways regulating protein synthesis[15]. In spite of this concomitant inhibition of the other AMPK downstream targets, the OGA inhibitor NButGT remains capable of fully blocking the anti-hypertrophic action of AMPK activator. This strongly suggests that O-GlcNAc inhibition remains the main mechanism by which AMPK blocks cardiac hypertrophy.

We identified two molecular mechanisms that explain how AMPK regulates HBP and O-GlcNAcylation during cardiac hypertrophy development. First, AMPK increases GFAT phosphorylation at Ser243. This AMPK phosphorylation site has been previously reported to decrease GFAT activity in different cellular models[35, 55]. GFAT being the rate limiting step of HBP, its inhibition by AMPK has great consequence in HBP and O-GlcNAcylation downstream events. In addition to GFAT regulation, we also ascertained a decrease in OGT protein expression after AMPK activation in our in vivo model of cardiac

hypertrophy. We ruled out a possible modification in Ogt mRNA (Supplementary Fig. 12). We could thus hypothesize that AMPK regulates OGT protein stability and consequently its expression. Supporting our data, a possible OGT phosphorylation by AMPK has been demonstrated[54]. It has to be mentioned that this AMPK-mediated OGT regulation does not seem mandatory. Indeed, OGT protein level was modified neither by PE nor by A769662 in our in vitro model of NRVMs (Supplementary Fig. 13). We postulate that OGT regulation requires more time in vivo than in our cellular model (24 h). Finally, we showed that, independently of hypertrophy, OGA protein level is modified in AMPK deficient mice. However, it is difficult to know if such AMPK/OGA regulation exists naturally in actual pathophysiological conditions.

By regulating major enzymes involved in O-GlcNAc signaling, AMPK modulates global protein O-GlcNAcylation levels. Several proteins involved in cardiac hypertrophy development have already been described as undergoing O-GlcNAcylation. Among them, we can for instance name the transcription factor c-Myc[57, 58] and the contractile components troponin I[59] and TnT[44]. Here, we demonstrated that TnT is one of the downstream targets of this AMPK-mediated regulation of O-GlcNAcylation. We are tempted to speculate that, by regulating global HBP, AMPK prevents O-GlcNAcylation processes linked to the activity of many other proteins. We can hypothesize that another mechanism controlled by AMPK and O-GlcNAcylation is autophagy. Indeed, our study revealed that AMPK activation might prevent hypertrophy in the face of ongoing elevated protein synthesis. Increasing autophagy is certainly a suitable way to counteract this increase in protein synthesis. Beclin1 and Bcl2 have been shown to be O-GlcNAcylated in diabetic hearts submitted to TAC, impairing autophagy[60]. We can presume that such proteins can be targeted by AMPK via O-GlcNAcylation regulation, even if AMPK can also directly promote autophagy[61]. In addition, we cannot exclude direct additional specific targeting of pro-hypertrophic mediators by AMPK, where O-GlcNAcylation and phosphorylation can compete for the same Ser/Thr site[62].

In summary, our study clearly highlights that AMPK inhibits cardiac hypertrophy mainly, if not solely, by regulating HBP and O-GlcNAcylation processes (Fig. 10g). This work provides novel insights into mechanisms linking cardiac hypertrophy and AMPK, with the emergence of O-GlcNAcylation as a novel putative therapeutic target. Additional studies are required to identify all the O-GlcNAcylated proteins involved in this phenomenon.

## Methods

**Animal care**. This study was approved by the Animal Research Committee of the Université catholique de Louvain and conformed to the American Heart Association Guidelines for Use of Animal in Research. All animals, housed with a 12 h/ 12 h light/dark cycle, had free access to water and standard chow (except if specified otherwise).

**Antibodies reagents and chemicals**. All antibodies used in the present study are listed in the Supplementary Table 2. AngII was from Merck. PE, metformin, PUGNAc, Aza, DON, AICAr, phenformin, and GlcN were from Sigma. Wheat germ agglutinin (WGA)/rhodamine was from Vector Biolab. A769662 was from R&D Systems.

**Preparation and treatment of NRVMs in primary culture**. NRVMs were isolated and cultured under aseptic conditions[63]. After sacrifice of 1- to 3-days old Wistar rat pups, hearts were harvested, placed in Hank's balanced salt solution (HBSS) and cut in 4 pieces. They were transferred into a 50ml-flask containing 30 ml of 1 g/L trypsin solution in HBSS and incubated 4 h at 4 °C under agitation. A second digestion was performed with a 0.5 g/L collagenase solution in Iscove's modified dulbecco's medium (IMDM). The digested hearts were centrifuged (1220 g, 5 min, 4 °C), resuspended in IMDM and placed on top of a 39% Percoll isotonic solution. Following centrifugation (3660 g, 30 min, 15 °C), cardiomyocytes were resuspended by pipetting them in IMDM containing 10% foetal bovin serum (FBS) and 2%

penicillin-streptomycin. Cardiomyocytes were plated and maintained for 2 days in the presence of 10% FBS and 2% penicillin-streptomycin at 37 °C and 5% $CO_2$. Serum-free NRVMs at a confluence of 80% were then treated for 24 h with pharmacological agents (PE [20 μM] or AngII [100 nM]) to induce hypertrophy. Different concentrations of A769662, phenformin or AICAr were used to activate AMPK during different periods of time. Several pharmacological agents were used to study O-GlcNAc signaling. OGA inhibitor (PUGNAc [50 μM] and GlcN [5 mM]) were used to increase O-GlcNAc levels whereas two different GFAT inhibitors, Aza (5 μM) and DON (20 μM), were utilized to reduce this process. See figure legends for more details about NRVMs treatments. NRVMs were lysed at the end of treatments, after removing the IMDM medium, in buffer containing 50 mM HEPES (pH 7.5), 50 mM KF, 1 mM KPi, 5 mM EDTA, 5 mM EGTA, 15 mM β-mercaptoethanol, a standard protease inhibitor mixture (complete mini, Roche) and 1% (vol/vol) Triton. Lysates were centrifuged (10000 g, 15 min, 4 °C) and supernatants were stored at −80 °C.

**AMPK knockdown by siRNA transfection.** NRVMs were transfected (at a confluence of 40–50%) with either control non-targeting siRNA (ON-TARGETplus Non-targeting siRNA, D-001810-01 from GE Healthcare, 50 nM) or pooled siRNA targeting both AMPKα1 and AMPKα2 catalytic isoforms (AMPKα1/α2 siRNA) (ON-TARGETplus Smart pool siRNAs, L-100623, and L-091373 from GE Healthcare, 50 nM) using lipofectamine RNAimax transfect reagent (Invitrogen) according to the manufacturer's protocol. After 66 h of transfection, medium was replaced and cells were treated for 24 h with pharmacological agents (PE [20 μM] or AngII [100 nM]) to induce hypertrophy and with different concentrations of A769662 as described in figure legends.

**Evaluation of cardiomyocyte hypertrophy.** α-actinin immunofluorescence staining was used to evaluate NRVM size[12]. NRVMs were plated on coverslips at a confluence of 40% before treatments. After treatments, NRVMs were fixed in 4% paraformaldehyde, permeabilized with 0.2% triton X-100 and blocked with 0.1% bovine serum albumin (BSA). Cells were incubated 1 h with the anti-α actinin primary antibody at room temperature and then 1 h with an anti-mouse secondary antibody coupled to Alexa fluor 594. Cells were visualized with an Axioskop 40 microscope and images were captured at ×200 magnification. Cell size (>70 cells per sample) was determined using Image J software.

**Evaluation of protein synthesis.** Amino acid incorporation into proteins was evaluated by measuring the incorporation of [14C]-phenylalanine into proteins of cultured NRVMs[63]. [14C]-phenylalanine (Perkin Elmer) (1μCi/ml) was added to NRVMs at the same time than other treatment. At the end of the treatment, NRVMs were washed twice with ice cold phosphate–buffered saline (PBS) and lysed as described above. Proteins were precipitated during 20 min by adding trichloroacetic acid giving a final concentration of 10% (vol/vol). Pellets containing proteins were washed with 100 mM NaOH and then dissolved with formic acid. Incorporated radioactivity into proteins was counted using a liquid scintillation counter.

**NFAT nuclear translocation and transcriptional activity.** Nuclear translocation of NFATc3 was evaluated by immunostaining. NRVMs were fixed as described above. Cells were incubated 1 h with the anti-NFATc3 primary antibody at room temperature and then with an anti-goat secondary antibody coupled to Alexa fluor 488. Percentage of positive nuclei (150 cells per sample) was measured using an Axioskop40 microscope on images captured at ×200 magnification. NFAT-dependent transcriptional activity was evaluated using a NFAT-luciferase reporter transgene as described[11]. Briefly, myocytes were infected with an adenoviral construct containing NFAT-luciferase reporter gene with NFAT binding sites upstream of luciferase (Ad.NFAT-Luc-Promoter adenovirus from Seven Hills Bioreagents). An adenoviral construction expressing β-galactosidase was used as control. Infection was performed at the multiplicity of infection of 10. Cells were treated 24 h post-infection as described in the figure legend. At the end of the treatment, cells were harvested with the reporter lysis buffer supplied in the luciferase assay system kit (Promega). Luciferase activity was then measured using the Victor X4 Multilabel Plate Reader (Perkin-Elmer).

**Preparation and treatment of ARVMs in primary culture.** ARVMs were prepared from male Wistar rats[64]. Two hearts were perfused using a Langendorff system with a Ca++-free Krebs-Henseleit buffer containing 5 mM glucose, 2 mM pyruvate, and 10 mM HEPES (pH 7.4). Isolation of ARVMs was obtained by adding 0.2 mM Ca++, 1 mg/ml collagenase (Worthington), and 0.4% (wt/vol) BSA to the perfusate for 30 min. The hearts were then removed from the perfusion system and cut into small fragments. Ca++ was progressively reintroduced to reach a final concentration of 1 mM. The pellet containing ARVMs was filtered and resuspended in MEM medium (Sigma) containing 1.2 mM Ca++, 2.5% FBS, 1% penicillin-streptomycin, and 20 mM HEPES pH 7.6. ARVMs were equally distributed in dishes coated with laminin (Sigma) and stored in a 95% O2, 5% CO2 incubator at 37 °C for 2 h. Cells were incubated in serum-free medium for 24 h before stimulation with PE 100 μM to induce hypertrophy. Phase-contrast images

of cultured ARVMs were taken after 48 h treatment and cell surface areas were analyzed using the Image J software as described previously[37].

**AngII-induced cardiac hypertrophy.** Twelve to sixteen week-old C57BL/6N male AMPKα2 deficient mice (AMPKα2 KO) and their control littermates (AMPKα2 WT) were treated with continuous infusion of AngII (2 mg/kg/d), or saline solution (for controls), for 5 or 14 days via osmotic mini-pumps (Alzet, 2002), which were subcutaneously implanted between the scapulae in anesthetized mice with Isoflurane. After treatment, animals were weighed before sacrifice under anesthesia induced by intraperitoneal injection of a solution (1 μl/g) of ketamine (100 mg/ml) and xylazine (20 mg/ml). Hearts were washed in PBS, weighed and then freeze-clamped with metal tongs chilled in liquid nitrogen and stored at −80 °C or fixed in 4% paraformaldehyde and embedded in paraffin for further analyses. For immunoblotting, hearts were immediately washed with PBS during 2 s to remove blood. Hearts were then freeze-clamped in liquid nitrogen and stored at −80 °C. Twenty milligram of heart was homogenized in 200 μl of RIPA lysis buffer (Thermo Scientific) supplemented with a protease/phosphatase inhibitor cocktail as well as O-GlcNAc inhibitors (1 μM Alloxan and 1 μM PUGNAc).

**TAC-induced hypertrophy.** Twelve to sixteen week-old WT and AMPKα2 KO mice (C57BL/6 N, male) underwent TAC to induce cardiac hypertrophy[65, 66]. Mice were anesthetized with an intraperitoneal injection of a mixture of ketamine (150 mg/kg) and xylazine (10 mg/kg) and placed in supine position. Under a microscope, an incision was performed on the right side of the sternum in order to avoid injury to the mammary artery. The thymus was pushed aside, and a constrictive band was placed and tightened around the aorta and a 27-G blunted needle, which was then removed. In sham-operated mice, the ligature was not tightened. Chest was closed with a polypropylene suture. Animals were treated with 0.1 mg/kg of buprenorphine after surgery. Doppler measurements of trans-stenotic gradients were systematically performed at day 1 and 3 weeks post-surgery. Only TAC mice with a velocity higher than 2.5 m/s were kept into experiment.

**Blood pressure telemetry recording in vivo.** Miniaturized implants (Data Sciences International) were surgically inserted in the aortic arch leading to measurement of blood pressure (BP) signals in conscious and unrestrained animals[67]. After 10 days of recovery, BP signals were measured (Ponemah v5.20) during 24 h as baseline data. Then, measurements were repeated after 2 weeks of AngII and metformin treatments in the same mice.

**Cardiac AMPK activation in vivo.** Male AMPKα2 KO and WT mice were treated daily with metformin (200 mg/kg/d) in drinking water.

**In vivo pharmacological increase in O-GlcNAc levels.** Mice were treated for 14 days with NButGT (50 mg/kg/d) by intraperitoneal injection (once a day). According to the increase in O-GlcNAc levels seen after 6 h of treatment, the last injection was done 6 h before sacrifice.

**Echocardiographic measurements.** Two-dimensional echocardiography was performed with a Vevo 2100 Imaging system (VisualSonics, Toronto, ON, Canada)[68]. Mice were anesthetized by isoflurane inhalation at concentrations between 4 and 4.5% (induction) and between 1.5 and 2% (maintenance), in 100% oxygen. LV mass was assessed from long axis images of the LV obtained at the level of papillary muscles. LV internal dimensions as well as the anterior and posterior wall thickness were measured at end-diastole. LV mass was computed using the following equation: LV mass = $1.05[(IVS + LVID + LVPW)^3 - (LVID)^3]$ where IVS is the interventricular septal thickness, LVID, the LV internal diameter and LVPW, the LV posterior wall thickness. LV mass was normalized by the tibia length measured at sacrifice. Measurements were performed by the same operator, blinded to the experimental groups.

**In vivo cardiomyocyte cross sectional area measurement.** Cardiomyocyte membranes were stained on heart sections deparaffinized and rehydrated. Slices were incubated 2 h with WGA/rhodamine (1/150) at room temperature before washing with PBS and mounting (vectashield H-1000, Vector Laboratories). Pictures were acquired with a Zeiss AxioImager microscope with a 40× objective. Cell size (>70 cells per sample) was determined using Axiovision software.

**Immunoblot analysis.** For immunoblotting, lysate supernatants (15–50 μg of total protein extract) were subjected to sodium dodecyl sulfate polyacrylamide gel electrophoresis and transferred onto polyvinylidenedifluoride membrane. Membranes were then probed with the appropriate antibodies to assess phosphorylation state or total protein level. The appropriate secondary antibody conjugated to HRP and the BM chemiluminescence blotting system (Roche) were used for detection. For O-GlcNAc detection in mouse heart lysates, two different protocols were utilized. The first protocol (concerning the main figures of the manuscript), RL2 antibody directly conjugated to HRP was used to prevent the detection of residual endogenous immunoglobulins if using a secondary antibody targeting mouse IgGs.

Twenty microgram of protein were loaded on a SDS-PAGE gel. After blocking, the membrane was incubated 1 h at room temperature with RL2-HRP antibody. The data obtained were confirmed by a second method (data presented in Supplementary Figure 7). In this protocol, lysates were first precleared with protein G sepharose (PGS) in order to fully remove endogenous immunoglobulins. For this preclearing, lysates (140 μg of protein) were incubated with 5 μl of PGS for 1 h before centrifugation. Twenty microgram of proteins of the resulting supernatants were then used for immunoblotting with RL2 and goat anti-mouse secondary antibody conjugated to HRP.

Band intensities were quantified by scanning and processing with program Image J. Band intensities obtained were normalized relative to a loading control (anti-eEF2 or anti-GAPDH, performed on the same gel). For anti-phospho antibodies, a second normalization was performed using their respective anti-total antibody (on another gel). When present, vertical lines denote that samples were run on the same gel but are not contiguous.

The uncropped version of all the blots can be found in Supplementary Figs 14 and 15.

**Gene expression analysis**. Total RNA was isolated using RNeasy mini-kit following the manufacturer's standard protocol (Qiagen). Reverse transcription was performed for 1 h at 37 °C with 1 μg of total RNA in the presence of oligo(dT) primers (primer poly(dT) (Roche) and Moloney-Murine Leukemia Virus Reverse Transcriptase (Invitrogen). Quantitative PCR was performed on IQ5 (Bio-Rad) using the QPCR Core kit for Sybergreen (Eurogentec). Levels of selected gene transcripts for each sample (Nppb; sense: gatctcctgaaggtgctgtcc, antisense: atccggtctatcttgtgccca; Ogt; sense: cctgggtcgcttggaaga, antisens: tggttgcgtctcaattgcttt), were averaged and normalized to the housekeeping gene, namely, hypoxanthine guanine phosphoribosyl transferase 1 (Hprt1; sense: ccagcgtcgtgattagcg, antisense: agcaagtctttcagtcctgtc), after ΔΔCt correction.

**Statistical analysis**. The sample size was not pre-determined by statistical analyses and was chosen according to previous publications. The 'N' number represents the number of animals for in vivo experiments and the number of biological replicates for in vitro experiment. No particular inclusion/exclusion criteria were applied. No particular randomization method was applied in this study. Quantification of cardiomyocyte area and echocardiography data analyses were performed by blinded researchers, the others experiments were performed in a non-blinded manner.

Results are expressed as mean ± s.e.m. unless otherwise stated. The statistical significance was calculated using unpaired Student's $t$-test or a One-way and Two-way analysis of variance using the Bonferroni post-hoc test. The significance threshold was set at $p < 0.05$.

**Data availability**. All relevant data are available from the authors upon reasonable request.

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

## Acknowledgements

The authors thank Louis Hue for his constant encouragement, helpful advice and critical reading of the manuscript. They also thank Cora Weigert (University of Tuebingen) for her technical help for GFAT phospho immunoblot assay. The authors wish to thank Servier Medical Art (http://smart.servier.com) for freely providing some of the illustrations used in this paper. This work was supported by grants from Fonds National de la Recherche Scientifique et Médicale (FNRS), Belgium, by Action de Recherche Concertée de la Communauté Wallonie-Bruxelles, Belgium (ARC11-16/035), by a PHC Tournesol from Communauté Wallonie-Bruxelles, Belgium, and by unrestricted grants from Astra Zeneca. F.M. and B.D. were supported by the Fund for Scientific Research in Industry and Agriculture, Belgium. R.G. received funding support from Institut de Recherche Expérimentale et Clinique, UCL. S.H. is Research Associate and L.B. is Senior Research Associate of FNRS, Belgium. C.B. is Postdoctorate Clinical Master Specialist of the FNRS, Belgium.

## Author contributions

R.G., F.M. performed most of the experimental work; J.D. and L.Bu. helped with experimental work; B.D. and A.G. mainly performed A769662 study when used at maximal concentration; E.D.-D. helped to prepare and incubate NRVMs; E.P.D. undertook echocardiography and related analysis; H.E., B.L., A.K.O., and B.B. provided help for surgery and in vivo experiments; B.V. generated AMPKα2 KO mice; K.S. provided help in the A769662 and GFAT experiments; L.Bu., C.G., B.V., K.S., C.D.R., J.-L.B., J.L.V., C.B., S.H. along with R.G., F.M., and L.Be. participated in study conception and design as well as data analysis and interpretation; R.G., F.M., and L.Be. drafted the manuscript; F.M., J.D., L.Bu., and L.Be. edited the final version; All co-authors revised the final manuscript version.

## Additional information

**Competing interests:** The authors declare no competing financial interests.

