## [Peer Review File · Nature Communications]

Reviewers' comments:

Reviewer #1

Remarks to the Author:

The manuscript reevaluates the role of AMPK activation in cardiomyocyte hypertrophy. Previous work indicated AMPK activation attenuates cardiomyocyte hypertrophy, and other studies indicated that cardiomyocyte hypertrophy required [an increase] in O-GlcNAcylation. The present study's contribution is to connect the two observations to one another, i.e., that AMPK activation inhibits O-GlcNAcylation to limit cardiomyocyte hypertrophy. The manuscript contains an exceptional amount of convergent data; the paper is logically laid out; and, the writing is sufficiently clear. The authors may want to consider the following issues:

Major:

1. The manuscript (page 4) does not accurately reflect the literature on the specific topic of HBP/O-GlcNAc in cardiomyocyte hypertrophy. Papers directly addressing this question are not mentioned, but should be; other relevant studies related to O-GlcNAc and heart failure are simply ignored. Elsewhere, there is misquoting of papers, particularly—for example—ref 24, which does not address O-GlcNAcylation.
2. Figure 4:
 - a. These data had the potential to show that AMPK activation blocks the relatively rapid induction of O-GlcNAcylation (see panel C). Unfortunately, such data are not provided. It remains possible that changes in hypertrophic signaling are occurring prior to the measured changes in O-GlcNAcylation. In other words, the change in O-GlcNAcylation may still be a coincidental finding, unless the AMPK activator blocks/blunts the effects reported in panel C. Such data are simple to acquire and do much more to support the manuscript's central theme.
 - b. Additional interpretation is needed for panel m. According to the data thus far, AMPK activation inhibits GFAT (presumably through phosphorylation) and this results in limitation of HBP flux. Such reduction in HBP flux then limits O-GlcNAcylation. If so, how does giving an OGA inhibitor (e.g., PUGNAc) work—if, after all, HBP flux was limited by AMPK-mediated inhibition of GFAT? The same question holds for the subsequent in vivo studies in which OGA inhibitors were used.

Minor:

1. For the mouse hearts: please provide details regarding the protein isolation/handling. Some of the most popular O-GlcNAc antibodies are raised in mice, which can create issues when using these antibodies in mouse tissues.
2. Figure 2: The major point of this figure is the differences in cardiomyocyte area. Considering this focus, it is not clear why there is a two-fold difference in area between panels B and K/L (and/or compare K/L to values Figures 1, 3, or 4).
3. Figure 1i: Typically, pERK and totalERK immunoblot data are generated by re-probing the same blot. However, this does not seem to have occurred because the bands are shaped differently. How can this be? The same is true for the p70S6K blots here, and for the ERK immunoblots throughout the manuscript.
4. It is not clear that Figure 1 contains information significantly different from that which can be found elsewhere in the paper. The authors may consider moving this figure to the supplement.
5. Line 557: An OGA inhibitor should not be equated with "HBP activation".
6. Why does the O-GlcNAc immunoblot in Figure 6 differ so much from the one shown in Figure 5?

Reviewer #2

Remarks to the Author:

This rather comprehensive and well-executed study demonstrates a role for O-GlcNAcylation in cardiac hypertrophy, and suppression of O-GlcNAcylation and hypertrophy by AMPK activation. They show that activation of AMPK with drugs at low levels suppress this PTM and hypertrophy at levels that do not have significant effects on other pathways such as NFAT and p70S6K. Interestingly, they show anti-hypertrophic effect in NRVM despite no suppression of protein synthesis. This observation argues for AMPK-mediated upregulation of degradation (perhaps by autophagy), and my only substantive criticism of the study is a failure to address this question of how does AMPK prevent hypertrophy in the face of ongoing elevated protein synthesis. However, given that this is a rather large study encompassing NRVM and in vivo studies, that question could be addressed in a subsequent manuscript. But it would be appropriate to address this with additional verbiage in discussion (autophagy is briefly mentioned but not in context with the protein synthesis results). The statement at line 167 (...without acting on well-documented AMPK targets, a still unidentified mechanism being highly suspected) omits consideration of autophagy as a major target of AMPK.

Minor comments:

Some western blots show intra-group variability or very modest intergroup differences; unfortunately, these are the ones that the authors neglected to provide quantitation, yet in text, they draw conclusions from those western blots.

There are several sentences with garbled syntax--it should be edited by someone with grammar skills.

Reviewer #3

Remarks to the Author:

This is a very interesting study that presents evidence that AMPK regulates cardiac hypertrophy via a pathway different from what has been published to date. I feel that the study can be improved by addressing the minor issues below:

Figure 1: Essentially a reproduction of existing findings with AMPK activation. Fine.

Figure 2: It appears as though the entire premise of this finding hinges on the sub-maximal activation of AMPK vs the maximal activation. However, it is not clear to me how sub-maximal AMPK activation was determined? Was it simply via the P-ACC blots or was it measured via the kinase assay?

Are the levels of AMPK expression altered in the presence of PE? If so, the same degree of inactivation using the siRNA may not be observed. Can the authors include the degree to which AMPK is inhibited in the presence of PE?

Figure 3: Why was AMPK activation not measured using the kinase assay? Only P-ACC is shown and this may not give a complete picture of AMPK activation.

Figure 4: Compelling data are presented but the figure is very difficult to follow. Also, what does the OGA inhibitor NButGT do to the other targets of AMPK such as p70, ERK, etc?

Figure 5: The quantification of O-GlcNAc and the blot of OGA are less than convincing. Are there other ways to determine the extent of O-GlcNAc? Do the authors have a positive control for OGA? What happens to AMPK activity in this situation? Also, are the hearts larger in this model given that O-GlcNAc is ostensibly increased and can stimulate cardiomyocyte growth?

Figure 6: What happens to AMPK activity in this situation? Also, based on the author's claims, that

elevated O-GlcNAc causes hypertrophy (and based on figure 6 d showing the highest amount of high MW O-GlcNAc) one would predict that these WT hearts are the most hypertrophic. Are they? They should be if O-GlcNAc is the key to hypertrophy.

Figure 7: What happens to AMPK activity in this situation?

Figure 9: The authors show AMPK activation using pAMPK in this figure. This is inconsistent with all of the other figures. Can pAMPK and pACC be shown for all figures?

Figure 10: The changes on OGT are less than impressive. Can the authors show more definitive data for panel 'a'?

The authors conclude that an "increase in protein O-GlcNAcylation is required for cardiomyocyte hypertrophy." If so, this should be the case in all models of hypertrophy. Is this the case? Again, if so, this would mean that any inhibitor of O-GlcNAcylation should be the best drug to inhibit cardiac hypertrophy. I would need to see these data to be convinced. Until then, the authors may want to temper their conclusions.

Response to the Reviewer's comments

We thank all the reviewers for their thorough review of the manuscript. With these insightful comments, we prepared a new version of our manuscript that, we think, greatly improves the impact of our work. Our responses to the reviewer's comments are detailed below.

Reviewer #1

General comment: “The manuscript reevaluates the role of AMPK activation in cardiomyocyte hypertrophy.... The manuscript contains an exceptional amount of convergent data; the paper is logically laid out; and, the writing is sufficiently clear.”

We thank the reviewer for the positive general comment.

Specific concerns:

Major concern 1: “The manuscript (page 4) does not accurately reflect the literature on the specific topic of HBP/O-GlcNAc in cardiomyocyte hypertrophy. Papers directly addressing this question are not mentioned, but should be; other relevant studies related to O-GlcNAc and heart failure are simply ignored. Elsewhere, there is misquoting of papers, particularly—for example—ref 24, which does not address O-GlcNAcylation.”

We thank the reviewer for his/her comment. The relationship between HBP/O-GlcNAc and cardiomyocyte hypertrophy is rather complex, characterized by different effects of O-GlcNAc depending on the context of cardiac hypertrophy. Indeed, O-GlcNAc plays distinctive roles in hypertrophy development, whether it is linked to diabetes (the main part of literature), or to physiological (exercise) or to more classical pathological (hypertensive) conditions. We recently published a review depicting this complex relationship (Mailleux et al, BBA 2016). In our manuscript, we focused on references linked to the topic of our study (classical hypertension-mediated pathological hypertrophy). We also referred to our review to point out this complex relationship (ref. 25 in version 1, now ref. 22). We did not extend this part of the introduction mainly due

to the size limit in reference number authorized by the editorial policies (maximum 70 references, which was the number of references we had in the first version). Nevertheless, we understand the reviewer's point and, now, extend this section by adding several important references linked to the topic (and removing other references elsewhere to respect the size limit of 70). Concerning studies related to O-GlcNAc and heart failure, we already referred, in the first version, to the study that concerns Troponin T (ref. 46 in version 1, now ref. 44). We now add the important paper of Jones' group published after the submission of our manuscript (Dassanayaka et al., Basic Res Cardiol, 2017) as well as their review published in 2014:

Page 4 “HBP is involved in multiple physiological processes but is also associated with undesirable cellular events in chronic diseases, such as diabetes inducing adverse effects in the heart (see 18,19 for review). In relation to cardiac pathologies, O-GlcNAcylation levels are increased during acute myocardial ischemia and chronic heart failure, but in these cases, with a cardioprotective effect^{18,20,21}. The role of O-GlcNAc during cardiac hypertrophy development is complex and still remains partly unclear^{18,21}. Action of O-GlcNAc largely depends of the context of cardiac hypertrophy with distinctive roles in hypertrophy development when linked to diabetes or to physiological exercise or to pressure overload pathological conditions^{18,21}. Regarding our topic, cardiac O-GlcNAc signaling and O-GlcNAcylation levels are increased in rat with pressure overload-mediated cardiac hypertrophy and in patients with aortic stenosis^{22,23}. Similarly, O-GlcNAc is increased in neonatal rat ventricular myocytes (NRVMs) submitted to pro-hypertrophic stimuli and pharmacological inhibition of O-GlcNAc signaling reverses the hypertrophic transcriptional reprogramming²³.”

And

Page 19 “In contrast to these in vitro data, it has to be mentioned that a very recent study performed by the same research group and using inducible/cardio-specific OGT-deficient mice submitted to TAC surgery revealed that OGT does not appear to be essential for cardiac hypertrophy development²⁰. However, the inducible animal model used in this study, the classical α -MHC-driven mutated estrogen

receptor-flanked Cre recombinase, allowed only partial OGT deletion²⁰. It is tempting to hypothesize that such partial deletion/reduction is not sufficient to have a significant impact on hypertrophy development. Nevertheless, further research will be necessary to definitively conclude on that issue. Even if required at least in vitro, O-GlcNAcylation does not seem sufficient to promote hypertrophy per se...

Finally, concerning ref. 24 (in version 1, now ref. 22), this study of Lunde and colleagues clearly addressed O-GlcNAcylation (O-GlcNAc levels as well as OGT, OGA and GFAT expression level) in *in vivo* models (mice and human) of hypertension-mediated pathological cardiac hypertrophy.

Major concern 2a: “Figure 4: These data had the potential to show that AMPK activation blocks the relatively rapid induction of O-GlcNAcylation (see panel C). Unfortunately, such data are not provided. It remains possible that changes in hypertrophic signaling are occurring prior to the measured changes in O-GlcNAcylation. In other words, the change in O-GlcNAcylation may still be a coincidental finding, unless the AMPK activator blocks/blunts the effects reported in panel C. Such data are simple to acquire and do much more to support the manuscript’s central theme.”

We thank the reviewer for his/her excellent suggestion. We performed such experiments that are now presented in additional panels (K and L) of the figure (Fig.3 in the new version). We concomitantly evaluated cardiomyocyte size and O-GlcNAc levels in a time-course from 2h to 24h in the presence or absence of A769662. The data obtained perfectly match our previous conclusion. Indeed, changes in hypertrophy is occurring simultaneously (and not prior) to the measured changes in O-GlcNAc levels.

Independently of the original question of the reviewer, this experiment brings additional interesting information. Indeed, AMPK activation does not prevent the initial increase in O-GlcNAc levels and cardiomyocyte size (from 2 to 8h), but exerts an inhibitory action later on. This reveals that there is a delay between the AMPK-mediated GFAT phosphorylation and inhibition (that occurs after 1h) and its action on O-GlcNAc levels (that decrease only after 8h). We can reasonably hypothesize that this delay is due to the time necessary for GFAT inhibition to have a significant impact on HBP flux and more generally on UDP-GlcNAc level. This hypothesis is in agreement with the recent

literature (from our collaborator and co-author, Prof. Sakamoto, Ref. 35) showing that the AMPK-mediated phosphorylation of GFAT induces only partial GFAT inhibition (see next concern). These data are summarized and discussed in the new version:

Page 9 “Remarkably, AMPK activation by 12.5 μ M A769662 blunted the increase in O-GlcNAc levels that occurs 24 h after PE treatment (Fig. 3f-i). Inasmuch as it has been recently shown that AMPK can phosphorylate GFAT on Ser243 inducing its inactivation³⁵, we evaluated the impact of 12.5 μ M A769662 on GFAT Ser243 phosphorylation and found a significant increase after 1 h of A769662 incubation (Fig. 3j). In agreement with the partial inhibition mediated by AMPK on GFAT³⁵, the effect of A769662 on cardiomyocyte size and O-GlcNAc levels were only observable after 8 h of treatment (Fig. 3k-l). We can reasonably hypothesize that this delay is due to the time necessary for GFAT inhibition to blunt HBP pathway. Nevertheless, those results show a perfect correlation between cardiomyocyte size and O-GlcNAc levels, reinforcing the link between O-GlcNAc increased and hypertrophy development.”

And

Page 19 “Second, O-GlcNAc levels perfectly correlate to cardiomyocyte size in all our experiments.”

[Redacted]

[Redacted]

Major concern 2b: “Additional interpretation is needed for panel m. According to the data thus far, AMPK activation inhibits GFAT (presumably through phosphorylation) and this results in limitation of HBP flux. Such reduction in HBP flux then limits O-GlcNAcylation. If so, how does giving an OGA inhibitor (e.g., PUGNAc) work—if, after all, HBP flux was limited by AMPK-mediated inhibition of GFAT? The same question holds for the subsequent in vivo studies in which OGA inhibitors were used.

The reviewer raised an interesting point. As explained in the previous paragraph, GFAT inhibition induced by AMPK phosphorylation is only partial (see figures below taken from Zibrova et al, 2016).

Left: GFAT activity in cells treated with AMPK activator (AICAR, left panel) compared to cells treated with DON or in absence of glutamine (Right panel) (From Zibrova et al., 2016).

In other words, AMPK-mediated inhibition of GFAT decreases but does not completely dampen down HBP flux. This explains the remaining action of OGA inhibitors. These compounds, by fully blocking O-GlcNAc removal, are able to counteract the partial inhibition induced by AMPK activation. This counteracting action of OGA inhibitors was already shown in the first version of the manuscript on O-GlcNAc levels *in vivo* (Fig. 8 of the new version, NButGT effect). We add in the new version the increase in O-GlcNAc levels induced by PUGNAc *in vitro* (New Fig 4C). We modified the text accordingly:

Page 11 “GFAT inhibition by AMPK being partial³⁵, such inhibition should not fully blunt HBP flux. In agreement with this hypothesis, OGA inhibition by PUGNAc was able to increase O-GlcNAc levels even in the presence of A769662 (78±22% for PE+A769662 vs. 247±98% for PE+A769662+PUGNAc, relative to PE treatment, p≤0.05) (Fig. 4c). Supporting data obtained with glucosamine, PUGNAc nicely counteracted the anti-hypertrophic action of the three AMPK activators (Fig. 4d).”

Minor concern 1: “For the mouse hearts: please provide details regarding the protein isolation/handling. Some of the most popular OGlcNAc antibodies are raised in mice, which can create issues when using these antibodies in mouse tissues.”

We thank the reviewer for raising this crucial point. Hearts were rapidly taken after sacrifice, rapidly washed in PBS to eliminate blood before freeze-clamping and preservation in -80° . PBS washing was performed taking into account the fact that the secondary antibody could be raised in mice. However, we verified if our samples were not contaminated by residual mouse antibodies and, indeed, IgGs can be detected (using secondary antibody alone, see figure below). So, we decided to use two different techniques to verify that this contamination did not alter our conclusions for all our *in vivo* experiments. The first method uses a primary antibody directly coupled to HRP (RL2-HRP) and, so, bypasses the use of the secondary antibody. The second method is to perform a preclearing of our samples with PGS before gel loading and then use the classical RL2 + anti-mouse secondary antibody coupled to HRP. An example of such experiments is presented here:

Both RL2-HRP and PGS pre-clearing methods allow to eliminate signal due to residual mouse antibodies. WT and AMPK α 2 KO mice were treated with or without angiotensin II (AngII, 2 mg/kg/d) for 5 days. Whole hearts were precleared with protein G sepharose in order to prevent the interaction of the secondary antibody (produced in mouse) with endogenous immunoglobulins. Total heart lysate (140 μ g of protein) were incubated with 5 μ l of PGS for 1 hour. 20 μ g of proteins with or without pre-clearing were then used for immunoblotting. Representative immunoblot of protein O-GlcNAcylation level in WT mouse hearts using either, O-GlcNAc antibody, O-GlcNAc-HRP coupled antibody or secondary anti-mouse antibody. LC = loading control.

This figure shows that both preclearing or RL2-HRP methods can be equally used to eliminate the signal coming from residual endogenous mouse antibodies. This example also shows that the conclusion previously made remains valid: i) O-GlcNAc

levels are increased in AMPK KO vs. WT; ii) AngII treatment increases O-GlcNAc levels in WT but not in AMPK KO mouse hearts.

Similar data were obtained for all the blots previously proposed in the first version. We replaced all of them by new ones using the RL2-HRP method in the main figures (and performed new quantifications). We also added data obtained with preclearing in the new Supplementary Figure 6 to reinforce our main data. To summarize, all the conclusions made previously remain accurate.

We also modified the text in the method section accordingly:

Page 27 “For immunoblotting, hearts were immediately washed with PBS during 2 seconds to remove blood. Hearts were then freeze-clamped in liquid nitrogen and stored at -80°C. 20 mg of heart was homogenized in 200 µl of RIPA lysis buffer (Thermo Scientific) supplemented with a protease/phosphatase inhibitor cocktail as well as O-GlcNAc inhibitors (1 µM Alloxan and 1µM PUGNAc).”

And

Page 29 “For O-GlcNAc detection in mouse heart lysates, two different protocols were utilized. The first protocol (concerning the main figures of the manuscript), RL2 antibody directly conjugated to HRP was used to prevent the detection of residual endogenous immunoglobulins if using a secondary antibody targeting mouse IgGs. 20 µg of protein were loaded on a SDS-PAGE gel. After blocking, the membrane was incubated 1 hour at room temperature with RL2-HRP antibody. The data obtained were confirmed by a second method (data presented in Supplementary Figure 6). In this protocol, lysates were first precleared with protein G sepharose in order to fully remove endogenous immunoglobulins. For this preclearing, lysates (140 µg of protein) were incubated with 5 µl of PGS for 1 hour before centrifugation. 20 µg of proteins of the resulting supernatants were then used for immunoblotting with RL2 and goat anti-mouse secondary antibody conjugated to HRP.”

Minor concern 2: “Figure 2: The major point of this figure is the differences in cardiomyocyte area. Considering this focus, it is not clear why there is a two-

fold difference in area between panels B and K/L (and/or compare K/L to values Figures 1, 3, or 4)."

The data presented in figure 2K/L (which is, now, Figure 1 K/L) are correct. These data come from a siRNA experimental procedure which is different from all others. Indeed, in these experiments, NRMVs are cultured for a longer period of time before starting pro-hypertrophic treatments (PE or PE+A769662). As explained in the method section, these NRVMs are normally cultured then transfected for 66 hours (period necessary to switch off the AMPK signal) before starting the hypertrophy treatment. NRVMs, which are non-matured cells, continue to grow in culture. So, this extra time (almost three days) is sufficient to have an impact on basal cardiomyocyte area. We cannot exclude an additional effect of the transfection agent, which is the same in all conditions (we compared non-targeted siRNA vs. AMPK-targeted siRNA).

Minor concern 3: "Figure 1i: Typically, pERK and totalERK immunoblot data are generated by re-probing the same blot. However, this does not seem to have occurred because the bands are shaped differently. How can this be? The same is true for the p70S6K blots here, and for the ERK immunoblots throughout the manuscript."

In our view (as for most scientists in the field), stripping a blot to detect the total protein level after detecting the phosphorylation of the same protein or vice versa is not recommended. Indeed, stripping might falsify the results due to unequal removal of antibodies from the membrane. When phospho- and total antibodies are coming from the same animal species, you can re-probe the first primary antibody that is not fully removed (the secondary antibody being the same). Biases could also be present even if the two antibodies are coming from different species. If the two epitopes are close one to each other, the second signal can be artifactually decreased due to steric interaction between the two primary antibodies. This classically occurs with the total p70S6K antibody that gives a weaker signal after re-probing when the anti-phospho signal is elevated.

As explained in details in the method section, total- and phospho-immunoblots are performed with the same protein extract in two different gels in parallel. Band intensities were quantified, firstly normalized relative to the loading control (anti-eEF2 or anti-GAPDH, performed on the same gel) and, secondly normalized using their respective anti-total antibody (on another gel).

Minor concern 4: “It is not clear that Figure 1 contains information significantly different from that which can be found elsewhere in the paper. The authors may consider moving this figure to the supplement.”

We thank the reviewer for this advice. This figure summarized the impact of A769662 on AMPK signaling and hypertrophy when used at saturated concentration. We thought that starting by these data made the story easier to understand. However, we agree with the reviewer. We moved Figure 1 into Supplementary Figures in the new version.

Minor concern 5: “Line 557: An OGA inhibitor should not be equated with “HBP activation.”

We agree with the reviewer. We modified the text:

Page 28 “In vivo pharmacological increase in O-GlcNAc levels.”

Minor concern 6: “Why does the O-GlcNAc immunoblot in Figure 6 differ so much from the one shown in Figure 5?”

We performed these gels with different batches of RL2 antibody. This could explain the difference in signal. Nevertheless, taking into account the first minor comment of the present reviewer, we re-did all the blots from our *in vivo* studies. The blots presented in this new version reveal similar band profiles.

Reviewer #2

General comment: “This rather comprehensive and well-executed study demonstrates a role for O-GlcNAcylation in cardiac hypertrophy, and suppression of O-GlcNAcylation and hypertrophy by AMPK activation. They show that activation of AMPK with drugs at low levels suppress this PTM and hypertrophy at levels that do not have significant effects on other pathways such as NFAT and p70S6K. Interestingly, they show anti-hypertrophic effect in NRVM despite no suppression of protein synthesis.”

We thank the reviewer for the positive general comment.

Specific concerns:

Major concern 1: “This observation argues for AMPK-mediated upregulation of degradation (perhaps by autophagy), and my only substantive criticism of the study is a failure to address this question of how does AMPK prevent hypertrophy in the face of ongoing elevated protein synthesis. However, given that this is a rather large study encompassing NRVM and in vivo studies, that question could be addressed in a subsequent manuscript. But it would be appropriate to address this with additional verbiage in discussion (autophagy is briefly mentioned but not in context with the protein synthesis results). The statement at line 167 (...without acting on well-documented AMPK targets, a still unidentified mechanism being highly suspected) omits consideration of autophagy as a major target of AMPK.”

We thank the reviewer for raising this important point. We now discuss more deeply the relationship between AMPK and autophagy in the discussion section.

We modified the text accordingly:

Pages 8-9 “Collectively, these results support the idea that low doses of AMPK activators are capable of preventing cardiomyocyte hypertrophy without acting on the aforementioned AMPK targets, suggesting the involvement of another mechanism.”

And

Page 22 “We can hypothesize that another mechanism controlled by AMPK and O-GlcNAcylation is autophagy. Indeed, our study revealed that AMPK activation might prevent hypertrophy in the face of ongoing elevated protein synthesis. Increasing autophagy is certainly a suitable way to counteract this increase in protein synthesis. Beclin1 and Bcl2 have been shown to be O-GlcNAcylated in diabetic hearts submitted to TAC, impairing autophagy⁶⁰. We can presume that such proteins can be targeted by AMPK via O-GlcNAcylation regulation, even if AMPK can also directly promote autophagy⁶¹.”

Minor concern 1: “Some western blots show intra-group variability or very modest intergroup differences; unfortunately, these are the ones that the authors neglected to provide quantitation, yet in text, they draw conclusions from those western blots.”

We now provide quantification for all the main western blots presented in figures (via graph or fold increase above blots) or in the result section of the text. As an example, the reviewer can find quantification of blots presented in Figure 2 (previously, Fig. 3), in new Supplementary Figure 3.

Due to the lack of space, the quantification of several minor western blots in Figure 1 and Supplementary Figure 1 is not presented. The reviewers can see them below, showing significant differences:

High A769662 concentration induces AMPK and ACC phosphorylation at early time point and persists for at least 24h. (a-d) NRVMs were treated with A769662 (100 μ M) for increasing times. (a-b) Representative immunoblot and quantification of ACC^{Ser79} phosphorylation. N=3. (c-d) Representative immunoblot and quantification of AMPK^{Thr172} phosphorylation. N=3. Data in a, c are mean \pm s.e.m. The data were analyzed using One-way ANOVA followed by Bonferroni post-test. *p<0.05 vs untreated cells.

And

Low A769662 concentration induces ACC phosphorylation and tends to phosphorylate AMPK. NRVMs were treated with (open bars) or without (solid bars) phenylephrine (PE, 20 μM) in the presence or absence of increasing concentration of A769662 (from 12.5 to 100 μM) for 24 h. (a-b) Representative immunoblot and quantification of ACC^{Ser79} phosphorylation. N=8. (c-d) Representative immunoblot and quantification of AMPK^{Thr172} phosphorylation. N=3. Data in a-c are mean \pm s.e.m. The data were analyzed using Two-way ANOVA followed by Bonferroni post-test. * $p < 0.05$ vs untreated cells, # $p < 0.05$ vs PE-treated cells.

For this reviewer's interest, it has to be mentioned that A769662 is known to mainly stimulate AMPK activity via an allosteric mechanism (Refs 24 & 25). Its action on AMPK phosphorylation is minor and is significant only at high concentration. This is the reason why the best way to evaluate AMPK signaling downstream A769662 is to follow the phosphorylation state of its bona fide substrate, ACC. Nevertheless, we also provide AMPK phosphorylation here as requested by reviewer 3.

Minor concern 2: "There are several sentences with garbled syntax--it should be edited by someone with grammar skills."

We apologize for these grammatical errors. The first version was already edited by an American Editing Society. We now performed a second edition via an expert in English grammar. We hope that the present version will satisfy the reviewer.

Reviewer #3

General comment: "This is a very interesting study that presents evidence that AMPK regulates cardiac hypertrophy via a pathway different from what has been published to date. I feel that the study can be improved by addressing the minor issues below"

We thank the reviewer for the positive general comment.

Specific concerns:

Minor concern 1: “Figure 1: Essentially a reproduction of existing findings with AMPK activation.”

As proposed by reviewer 1, we moved this figure in Supplementary figures.

Minor concern 2: “Figure 2: It appears as though the entire premise of this finding hinges on the sub-maximal activation of AMPK vs the maximal activation. However, it is not clear to me how sub-maximal AMPK activation was determined? Was it simply via the P-ACC blots or was it measured via the kinase assay?”

Actually, the concentrations used were chosen based on previous studies performed in the lab (Timmermans et al, AJPHCP 2014 and de Meester et al., Cardiovasc Res 2014) but also in line with those used in the literature. In our previous studies performed in stem cells and adult cardiomyocytes, 100 μ M was shown to maximally activate AMPK and to induce maximal ACC phosphorylation whereas dose-response curves made by progressively decreasing the concentration revealed that 12.5 μ M was still able to activate AMPK signaling but at a lower level. This is the reason why we used the same concentrations in the present manuscript and why we choose the terms “maximal” and “sub-maximal”.

We now clarify this point by using “low” instead of “submaximal” and change the text at the beginning of the result section accordingly:

Page 6 “We started by using a concentration of 100 μ M giving maximal AMPK activation in adult cardiomyocytes²⁶. We showed that, at this concentration, A769662 rapidly increased AMPK activity, which was maximal at 5 min and persisted for at least 24 h (Supplementary Fig. 1a). The same applied for the phosphorylation of its bona fide substrate acetyl-CoA carboxylase (ACC) (Supplementary Fig. 1b).”

Minor concern 3: “Are the levels of AMPK expression altered in the presence of PE? If so, the same degree of inactivation using the siRNA may not be observed.

Can the authors include the degree to which AMPK is inhibited in the presence of PE?”

The reviewer raises an interesting point. As shown in the figures below, PE treatment does not significantly modify AMPK expression. Moreover, RNA silencing is equally efficient in control and PE treatments.

Phenylephrine treatment does not modify AMPKα expression. Neonatal cardiomyocytes were transfected with non-targeting siRNA and with siRNA directed against both AMPKα subunits for 66h and then treated with A769662 (12.5 μM) and phenylephrine (PE, 20 μM) for 24h. (a) Representative immunoblots of AMPKα protein expression at low and high exposure. (b) Quantification of AMPKα protein expression in basal conditions. N=8. (c) Quantification of AMPKα protein expression in non-targeting siRNA conditions and in AMPKα1/α2 siRNA conditions. N=4. Data in b-c are mean ± s.e.m. The data were analyzed using Two-way ANOVA followed by Bonferroni post-test in c and using unpaired Student's t-test in b. #p<0.05 vs. non-targeting control and PE conditions. C = control, A = A769662, PE = phenylephrine and PE+A = phenylephrine + A769662, LC = loading control.

Due to the high number of figures presented in the new version of the manuscript, we do not think that these data should be formally added. We just mention this aspect in the text as follows (However, if the reviewer regards the presentation of these data essential, we can of course add them in our Supplementary figures):

Page 8 “This AMPK deletion, which was similar in control and PE-treated cells (data not shown), abolished the anti-hypertrophic action of A769662 (Fig. 1k,l). This confirmed that A769662-mediated inhibition of NRVM hypertrophy is an AMPK-dependent mechanism.”

Minor concern 4: “Figure 3: Why was AMPK activation not measured using the kinase assay? Only P-ACC is shown and this may not give a complete picture of AMPK activation.”

The reviewer raises another interesting point. We now added AMPK phosphorylation data (New Fig. 2C and H). Similarly, both AMPK and ACC phosphorylation were shown in Fig. 1A.

We preferentially performed pAMPK (corresponding to *in situ* AMPK activation) paired to pACC (corresponding to *in situ* AMPK signaling stimulation) than *in vitro* kinase assay because this assay is performed in the presence of saturating concentration of AMP (which is necessary to evaluate AMPK activity). In other words, this *in vitro* situation does not correspond to *in situ* situation where free AMP concentration is unknown.

Minor concern 5: “Figure 4: Compelling data are presented but the figure is very difficult to follow. Also, what does the OGA inhibitor NButGT do to the other targets of AMPK such as p70, ERK, etc?”

Previous figure 4 has been split into two figures in the new version, Fig. 3 and 4. We hope that the reviewer will find this way to present easier to follow.

Concerning the effect of the OGA inhibitor NButGT on the other AMPK targets, we assume that the reviewer means other OGA inhibitor PUGNAc (which is used *in vitro* and presented in figure 4), whereas NbutGT is used *in vivo* later in the manuscript. As requested by the reviewer, we measured p70, ERK phosphorylation as well as NFAT activity under the OGA inhibitor PUGNAc treatment (presented in the figure below).

PUGNac has no effect on ERK1/2 and p70S6K phosphorylation but induces NFAT transcriptional activity. NRVMs were treated with (open bars) or without (solid bars) phenylephrine (PE, 20 μ M) in the presence or absence of A769662 (12.5 μ M) and with or without PUGNac (50 μ M) for 24 h except for ERK1/2 phosphorylation which has been evaluated after 1h. (a) Quantification of ERK1/2^{Thr202/Tyr204} phosphorylation. N=3. (b) Quantification of p70S6K^{Thr389} phosphorylation. N=4-9. (c) Evaluation of NFAT transcriptional activity by luciferase activity. N=4-10. Data in a-c are mean \pm s.e.m. The data were analyzed using Two-way ANOVA followed by Bonferroni post-test in a-b and One-way ANOVA followed by Bonferroni post-test in c. *p<0.05 vs. untreated cells.

PUGNac did not modify p70S6K or ERK phosphorylation state whatever the cell treatments (control, A769662, PE or PE+A769662). By contrast, PUGNac, alone, was able to increase NFAT activation. This last effect was rather expected because NFAT O-GlcNAcylation is known to occur and to induce its translocation into the nucleus (Facundo et al., AJP_{HCP} 2012). This point is interesting because PUGNac alone is not sufficient to induce cardiac hypertrophy (Figure 4), confirming that NFAT activation seems necessary but not sufficient for hypertrophy development.

Moreover, PUGNac did not increase NFAT response to PE or PE+A769662. In other words, the regulation of NFAT activity by PE and by A769662 is not modified by OGA inhibitor, definitively ruling out the possibility of a main role of NFAT in the action of AMPK activator at low concentration.

Even if all these data are interesting, we think that they are a bit outside the main scope of the present study. So, we think that it would be better not to add them in the manuscript, thereby avoiding the risk of diluting the main message.

Minor concern 6, Part 1: “Figure 5: The quantification of O-GlcNAc and the blot of OGA are less than convincing. Are there other ways to determine the extent of O-GlcNAc? Do the authors have a positive control for OGA?”

First, as explained in response to the minor concern 1 of Reviewer 1, we re-did all O-GlcNAc blots for the *in vivo* part of the manuscript using two different techniques (presented in new Fig. 5,6 and 8 as well as in the new Supplementary figure 6). We hope that this reviewer find these new blots and quantification more convincing.

Concerning OGA, we also present a new and more demonstrating blot (new Fig. 5A). Moreover, we checked the validity of the antibody used by performing western blots in NRVMs where OGA was knocked-down by siRNA (see below):

OGA protein expression is repressed in AMPKα2 KO mice and by siRNA directed against OGA in NRVMs: (a) low and high exposure immunoblots of OGA protein expression in WT and AMPKα2 KO mouse hearts. (b) Representative immunoblot of OGA protein expression in NRVMs transfected with non-targeting siRNA and siRNA directed against OGA. LC = loading control.

Minor concern 6, Part 2: “Figure 5: What happens to AMPK activity in this situation? Also, are the hearts larger in this model given that O-GlcNAc is ostensibly increased and can stimulate cardiomyocyte growth?”

First, AMPK activity in AMPK α 2 KO versus WT mouse heart has been thoroughly evaluated in a previous study of our group (Zarrinpashneh et al, AJP_{HCP}, 2006, 291(6):H2875-83). For this reviewer's interest, we showed in this study that AMPK α 2 activity is totally absent whereas AMPK α 1 activity is similar in both strains. AMPK α 1 remains normally expressed and active in the AMPK α 2 KO, demonstrating that AMPK α 2 plays the major role in the regulation of cardiac hypertrophy. We already extensively discussed this isoform specificity in the first part of the discussion section. Concerning the second question, even if O-GlcNAc is ostensibly increased under basal condition, the heart of AMPK α 2 KO is not larger than that of WT (see Supplementary Table 1 already present in the first version of the manuscript). This is in total agreement with two other observations presented here. First, PUGNAc and glucosamine were able to increase O-GlcNAc levels in control NRVMs but this did not induce hypertrophy per se (New Fig. 4). Second, the same applies *in vivo* where NButGT did not induce cardiac hypertrophy in control WT mice whereas it increased O-GlcNAc levels. In other words, the increase in O-GlcNAcylation is necessary but not sufficient to induce hypertrophy. We already extensively discussed this aspect in the discussion section:

Page 19 "Even if required at least in vitro, O-GlcNAcylation does not seem sufficient to promote hypertrophy per se. Indeed, glucosamine, PUGNAc (in vitro) and NButGT (in vivo) are unable to induce hypertrophy in the absence of pro-hypertrophic agents. The same applies for AMPK α 2 KO mice, which are characterized by increased levels of O-GlcNAc under basal state without any significant effect on heart size under basal conditions. This highly suggests that O-GlcNAcylation must be combined with other molecular events, such as the increase in protein synthesis and modifications in gene transcription to promote cardiac hypertrophy."

Minor concern 7: "Figure 6: What happens to AMPK activity in this situation? Also, based on the author's claims, that elevated OGlcNAc causes hypertrophy (and based on figure 6 d showing the highest amount of high MW O-GlcNAc) once would predict that these WT hearts are the most hypertrophic. Are they? They should be if O-GlcNAc is the key to hypertrophy."

To answer question 1, we performed western blot analysis of ACC phosphorylation to evaluate AMPK signaling activity in all conditions presented in figures 6 and 7. These data are now presented in Supplementary Figure 6. To summarize, ACC phosphorylation was induced by metformin in WT but not in AMPK α 2 KO under both control or AngII treatments (see also next concern).

Concerning question 2, we re-did all O-GlcNAc blots of our *in vivo* experiments (see Minor Concern 1 of Reviewer 1 for more details). The blot shown in figure 6 gives a clear answer. AngII elevates all O-GlcNAc bands equally in WT but not in AMPK α 2 KO mouse hearts, the latter being characterized by high levels of O-GlcNAcylation under basal condition.

As explained in our answer to the previous concern, simply correlating O-GlcNAc levels and degree of hypertrophy looks hazardous, O-GlcNAcylation being necessary but not sufficient to induce hypertrophy.

Minor concern 8: “Figure 7: What happens to AMPK activity in this situation?”

We performed western blot analysis of ACC phosphorylation to evaluate AMPK signaling activity in all conditions presented in figures 6 and 7. These data are presented in Supplementary Figure 6. To summarize, ACC phosphorylation was induced by metformin in WT but not in AMPK α 2 KO under both control or AngII treatments.

The text is modified accordingly:

Page 13 “As expected, metformin efficiently increased AMPK signaling (followed by ACC phosphorylation) in WT but not in AMPK α 2 KO hearts (Supplementary Fig. 7).”

Minor concern 9: “Figure 9: The authors show AMPK activation using pAMPK in this figure. This is inconsistent with all of the other figures. Can pAMPK and pACC be shown for all figures?”

In addition to pAMPK, we now add pACC, in Figure 9. Both drive to a similar conclusion: phosphorylation is increased under metformin treatment in WT but not in AMPK α 2 KO mouse hearts.

Moreover, both pACC and pAMPK data are now presented in all our main figures (See also Minor Concern 1 of Reviewer 2 for more details).

Minor concern 10: “Figure 10: The changes on OGT are less than impressive. Can the authors show more definitive data for panel ‘a’?”

We performed new immunoblotting experiments that confirmed a clear increase in OGT protein level after AngII treatment in WT hearts. We now present a new representative blot in panel ‘a’.

Minor concern 11: “The authors conclude that an “increase in protein O-GlcNAcylation is required for cardiomyocyte hypertrophy.” If so, this should be the case in all models of hypertrophy. Is this the case? Again, if so, this would mean that any inhibitor of O-GlcNAcylation should be the best drug to inhibit cardiac hypertrophy. I would need to see these data to be convinced. Until then, the authors may want to temper their conclusions.”

We thank the reviewer for his/her comment. As explained in the Major Concern 1 of Reviewer 1, the relationship between HBP/O-GlcNAc and cardiomyocyte hypertrophy is rather complex, characterized by different effects of O-GlcNAc depending on the context of cardiac hypertrophy. Indeed, O-GlcNAc plays distinctive roles in hypertrophy development, whether linked to diabetes (as illustrated in the main part of literature), or to physiological (exercise) or to more classical pathological conditions (e.g. hypertension). We recently published a review describing this complex relationship (Mailleux et al, BBA 2016). We focused our references on the topic of our study (classical hypertension-mediated pathological hypertrophy). We also referred to our review to point out this complex relationship (ref. 25 in version 1, now ref. 18).

We modified the text accordingly and also temper our conclusions as follows:

Page 4 “HBP is involved in multiple physiological processes but is also associated with undesirable cellular events in chronic diseases, such as diabetes inducing adverse effects in the heart (see 18,19 for review). In relation to cardiac pathologies, O-GlcNAcylation levels are increased during acute myocardial ischemia and chronic heart failure, but in these cases, with a cardioprotective effect^{18,20,21}. The role of O-GlcNAc during cardiac hypertrophy development is complex and still remains partly unclear^{18,21}. Action of O-GlcNAc depends of the context of cardiac hypertrophy with distinctive roles in hypertrophy development when linked to diabetes or to physiological exercise or to pressure overload

pathological conditions^{18,21}. Regarding this last condition, cardiac O-GlcNAc signaling and O-GlcNAcylation levels are increased in rat with pressure overload-mediated cardiac hypertrophy and in patients with aortic stenosis^{22,23}. Similarly, O-GlcNAc is increased in neonatal rat ventricular myocytes (NRVMs) submitted to pro-hypertrophic stimuli and pharmacological inhibition of O-GlcNAc signaling reverses the hypertrophic transcriptional reprogramming²³.”

And

Page 18-19 “... it appears from our data that O-GlcNAcylation elevation is not just a marker of cardiac hypertrophy but could also be considered as a required event in this pathological process. First, O-GlcNAcylation occurs early in the hypertrophic process (within 2h of pro-hypertrophic treatment in our in vitro NRVM model and within the first 5 days of in vivo treatment in AngII-treated mice). Second, O-GlcNAc levels perfectly correlate to cardiomyocyte size in all our experiments. Third, our in vitro study indicates that preventing this increase in O-GlcNAcylation by inhibiting GFAT via the use of DON or Aza is sufficient to block NRVM hypertrophy development. In agreement, it has been shown that GFAT inhibition by DON prevents PE-induced hypertrophic markers to progress in NRVMs²³. In contrast to these in vitro data, it has to be mentioned that a very recent study performed by the same research group and using inducible/cardio-specific OGT-deficient mice submitted to TAC surgery revealed that OGT does not appear to be essential for cardiac hypertrophy development²⁰. However, the inducible animal model used in this study, the classical α -MHC-driven mutated estrogen receptor-flanked Cre recombinase, allowed only partial OGT deletion²⁰. It is tempting to hypothesize that such partial deletion/reduction is not sufficient to have a significant impact on hypertrophy development. Nevertheless, further studies will be necessary to reach a definitive conclusion regarding this issue. Even if required at least in vitro, O-GlcNAcylation does not seem sufficient to promote hypertrophy per se...”

REVIEWERS' COMMENTS:

Reviewer #1 (Remarks to the Author):

The authors are credited with a comprehensive response to my previous comments. I have no additional criticisms of substance; however, the authors may want to mention the identity of their loading controls (aka, "LC") in the immunoblots. The methods state EF-2 or GAPDH. This is simply a suggestion for clarity for the reader; the authors'/editor's ultimate decision on this minor point does not affect my recommendation.

Reviewer #3 (Remarks to the Author):

The authors have addressed all of my previous concerns.

Response to the Reviewer's comments

We thank all three reviewers for their positive final review. Our response to the last reviewer's request is detailed below.

Reviewer #1

Remaining concern: "The authors are credited with a comprehensive response to my previous comments. I have no additional criticisms of substance; however, the authors may want to mention the identity of their loading controls (aka, "LC") in the immunoblots. The methods state EF-2 or GAPDH. This is simply a suggestion for clarity for the reader; the authors'/editor's ultimate decision on this minor point does not affect my recommendation."

We modified the figures and figure legends according to the suggestion of the reviewer. GAPDH and eEF-2 are, now, mentioned. For reviewer information, we used GAPDH (located in the lower part of the gel) when high molecular weight proteins were studied. Whereas we utilized eEF-2 (located in the upper part of the gel) when low molecular weight proteins were analyzed.